# Hepatocyte FBXW7-dependent activity of nutrient-sensing nuclear receptors controls systemic energy homeostasis and NASH progression in male mice

Hui Xia [1,2], Catherine R. Dufour [1], Younes Medkour [1], Charlotte Scholtes [1], Yonghong Chen[1,2], Christina Guluzian[1,2], Wafa B'chir[1] & Vincent Giguère [1,2] ✉

Nonalcoholic steatohepatitis (NASH) is epidemiologically associated with obesity and diabetes and can lead to liver cirrhosis and hepatocellular carcinoma if left untreated. The intricate signaling pathways that orchestrate hepatocyte energy metabolism and cellular stress, intrahepatic cell crosstalk, as well as interplay between peripheral tissues remain elusive and are crucial for the development of anti-NASH therapies. Herein, we reveal E3 ligase FBXW7 as a key factor regulating hepatic catabolism, stress responses, systemic energy homeostasis, and NASH pathogenesis with attenuated *FBXW7* expression as a feature of advanced NASH. Multiomics and pharmacological intervention showed that FBXW7 loss-of-function in hepatocytes disrupts a metabolic transcriptional axis conjointly controlled by the nutrient-sensing nuclear receptors ERRα and PPARα, resulting in suppression of fatty acid oxidation, elevated ER stress, apoptosis, immune infiltration, fibrogenesis, and ultimately NASH progression in male mice. These results provide the foundation for developing alternative strategies co-targeting ERRα and PPARα for the treatment of NASH.

Nonalcoholic fatty liver disease (NAFLD), primarily defined as excessive lipid accumulation in the liver, is the hepatic manifestation of metabolic disorders associated with the global increased rate of obesity and type 2 diabetes[1,2]. Nonalcoholic steatohepatitis (NASH) is a more severe form of NAFLD that is further characterized by chronic inflammation and fibrosis[3]. Liver injuries associated with NASH can lead to cirrhosis, the need for a liver transplant and to the initiation and progression of hepatocellular carcinoma (HCC)[4]. Several diverse and parallel cellular processes synergistically induce NASH progression from simple steatosis, including lipotoxicity, endoplasmic reticulum (ER) stress, mitochondrial dysfunction, autophagy, apoptosis, inflammatory response and fibrogenesis[5–7]. In addition, multiple intrahepatic cell types, extrahepatic organs as well as the immune system have been implicated in the development of NASH[5,6]. This multiplicity of factors results in the complex etiology of NASH and has led to the development of several distinct therapeutic approaches currently being tested in clinical trials, including diet intervention, but pharmaceutical medicines for the treatment of NAFLD or NASH have yet to be approved by the FDA[8–10]. Moreover, it remains unknown whether the etiology of NASH is the result of the combined actions of several independent factors or multiple converging pathways governed by a single class of master regulators of energy homeostasis that would be more amiable to medical intervention.

The ubiquitin-proteasome system (UPS) has previously been implicated in the etiology of NASH and controls many critical cellular processes via governing the expression of regulatory proteins. Liver-

[1]Goodman Cancer Institute, McGill University, Montréal, QC H3A 1A3, Canada. [2]Department of Biochemistry, Faculty of Medicine and Health Sciences, McGill University, Montréal, QC H3G 1Y6, Canada. ✉e-mail: vincent.giguere@mcgill.ca

specific deletion of the genes encoding the E3 ligases FBXW7 and TRIM16 trigger excessive hepatic deposition of triglyceride (TG) and steatohepatitis[11,12]. FBXW7 is of particular interest in the context of NASH as it is known to regulate the protein stability of key metabolic transcription factors including SREBP1/2, the nuclear receptors REV-ERBα (NR1D1) and ERRα (NR3B1) as well as the coregulator proteins NCOA3 and PGC-1α[13–18]. Elevated lipogenesis, thought to initiate NAFLD and contribute markedly to hepatic steatosis[19,20], has long been proposed to be triggered by overactivation of SREBP1, the major transcriptional regulator of lipid biosynthesis and a FBXW7 substrate[11,13]. However, a separate study indicates that elevated fatty acid (FA) uptake and TG synthesis regulated by a KLF5/PPARγ2 pathway rather than SREBP1-mediated FA synthesis are responsible for *Fbxw7*-depletion-induced NAFL[21]. These disagreements and limitations of currently published mechanisms highlight the necessity to further explore the contribution of FBXW7 to NASH pathogenesis using newly developed technologies. Hepatic FBXW7 activity may also have a broader impact on the control of liver energy metabolism and associated systemic regulatory communication between distinct organs that remain to be fully investigated.

Nuclear receptors play critical roles in the control of cellular energy homeostasis[22]. The activity of a significant number of nuclear receptors has been linked with NAFL/NASH development, and numerous therapeutic agents targeting nuclear receptors are currently in clinical trials for treatment[23]. Of interest, the nutrient-sensing nuclear receptors PPARα (NR1C1) and ERRα, while co-regulating diverse physiological processes through targeting overlapping gene sets involved in lipid metabolism, inflammation, and apoptosis[24–28], have been shown to have opposing impact on the development of NAFLD. Mice with whole-body or liver-specific ablation of *Ppara* are more susceptible to high-fat diet (HFD) or NASH diet-induced obesity and liver steatosis[29–31]. In contrast, ERRα-null mice are protected from HFD-induced NAFL[32]. Together, these observations suggest that PPARα and ERRα may act in concert to regulate hepatic energy metabolism and that perturbation of their transcriptional activity may break the balance of liver homeostasis and drive NAFL/NASH progression.

Here, we reveal a previously undefined role for Fbxw7 and its substrates in maintaining hepatic catabolism and systemic homeostasis. We demonstrate that the development of NASH in mice with hepatocyte-specific deletion of *Fbxw7* is driven by the constitutive expression and activation of ERRα. Mechanistically, enhanced ERRα levels promote its recruitment to PPARα regulatory regions in part via ERRα-mediated attenuation of PPARα transcription, revealing a previously unrecognized context-dependent competition for DNA binding between ERRα and PPARα. The upregulation of ERRα signaling and downregulation of PPARα activity contributes to the suppression of fatty acid oxidation (FAO), increased ER stress, apoptosis, and immune infiltration but independent of de novo lipogenesis (DNL). Activating Fbxw7 and inhibiting ERRα both relieve hepatic ER stress and apoptosis, attenuating NASH development. We also uncover ERRα as a transcriptional regulator of *Fbxw7* expression thus establishing a regulatory feedback loop, and further demonstrate the potential for pharmacological inhibition of uncontrolled ERRα activity to manage NASH.

## Results

### FBXW7 protects against NASH

To first reveal the clinical relevance of FBXW7 in human NASH progression, we compared *FBXW7* expression in patients with NAFL and different stages of NASH from published transcriptome datasets. This analysis showed that *FBXW7* declines as NASH progresses and is lower in late-stage liver fibrosis (Fig. 1a, b). Also, *FBXW7* expression tends to decrease in cirrhotic human livers and is significantly lower in HCC (Supplementary Fig. 1a, b). Consistently, low *FBXW7* expression is linked to a worse survival rate among patients with HCC

(Supplementary Fig. 1c). The contribution of FBXW7 to NASH and HCC prompted us to examine its biological role. We thus generated mice with hepatocyte-specific loss of Fbxw7 activity (*Fbxw7*[L−/−]) by breeding mice harboring a conditional *Fbxw7* allele with mice carrying an Alb-Cre recombinase transgene to excise exons encoding the F-box domain required for ubiquitination complex assembly, which resulted in more than 29% increase in liver weight in comparison with their *Flox* littermate controls (Supplementary Fig. 1d–g). Principal component analysis (PCA) of the RNA-seq data of control and Fbxw7-null livers showed a substantial difference between genotypes, as well as tight clustering across replicates, indicating low contamination and high reproducibility (Supplementary Fig. 1h). Transcriptome analysis revealed a 342-gene subset dysregulated in livers of advanced versus early-stage NASH patients that are also perturbed in *Fbxw7*[L−/−] mice (Supplementary Fig. 1i). GO cellular component analysis further demonstrated that these differentially expressed genes (DEGs) of NASH progression or Fbxw7-null livers were most enriched in "Collagen-containing extracellular matrix", followed by "Basement membrane", "Focal adhesion", and "Endoplasmic reticulum lumen" (Supplementary Fig. 1j), in agreement with previous findings that NASH is accompanied by extracellular matrix (ECM) accumulation largely formed by type I collagens together with prolonged ER stress[6,33,34]. RT-qPCR confirmed overall upregulation of a subset of NASH progression signature genes[35] in *Fbxw7*-null livers, including *Akr1b10*, *Ccl20*, *Clic6*, *Col1a1*, *Col1a2*, *Epb41l4a*, *Fermt1*, *Gdf15*, *Itgbl1*, *Ltbp2*, and *Thy1* (Fig. 1c). These genes are known to gradually increase upon NASH progression, with *Gdf15* and *Ccl20* encoding proinflammatory cytokines, *Col1a1* and *Col1a2* encoding collagen chain proteins, and *Itgbl1* known for regulating liver fibrosis. Inflammation and fibrosis are the hallmarks of NASH, distinguishing it from NAFL, with the signature genes being significantly enriched and upregulated in Fbxw7-null livers demonstrated by Gene Set Enrichment Analysis (GSEA) (Supplementary Fig. 1k, l). In accordance, Fbxw7-null livers showed significant elevation of genes encoding essential proinflammatory cytokines and chemokines (*Tnfa*, *Il1rn*, *Il1b*, *Il6*, *Ccl2*) as well as major pro-fibrogenic cytokines (*Tgfb2*, *Ccn2/Ctgf*), resulting in elevated expression of fibrogenic genes (*Acta2/αSMA*, *Col3a1*) (Fig. 1d). These findings were further validated at the protein level in both liver and serum (Fig. 1e). Deconvolution of liver gene expression profiles using CIBERSORT and EPIC algorithms also revealed that Fbxw7 deficiency greatly elevated the overall abundance of hepatic infiltrating immune cells attributed to the significant rise of intrahepatic macrophages and fibroblasts proportions (Fig. 1f, g). Consistently, scar-associated macrophages (SAMac)-defining signature genes[36] and hepatic stellate cell signature genes[37] were significantly enriched and upregulated in Fbxw7-null livers (Fig. 1h, i). Immunohistochemistry of macrophage marker Aif1 and hepatic stellate cell/fibroblast marker αSMA confirmed elevated immune infiltration of Fbxw7-null livers (Fig. 1j, k). Fbxw7-null livers also showed significant enrichment in a subset of genes characterizing the HCC-related progenitor cell signature[38] (Supplementary Fig. 1m), which could be the origin of liver cancer. Inflammatory responses in hepatocytes can arise from apoptotic signaling triggered by severe ER stress, which is important for NASH progression and the initiation of HCC. Fbxw7-depleted livers showed upregulation of an apoptosis gene signature (Supplementary Fig. 1n) and elevated mRNA expression of genes encoding markers of ER stress (*Ddit3/Chop*, *Atf3*, *Phlda3*) and apoptosis (*Bax*, *Apaf1*, *Casp12*) (Fig. 1l). ER overload upon Fbxw7 deficiency was further indicated by increased eIF2α phosphorylation and robustly elevated CHOP protein (Fig. 1m). Indeed, Fbxw7-null livers displayed elevated expression and cleavage of caspase-12, a specific modulator of ER stress-mediated apoptosis, and subsequent activation of the major executioner caspase (caspase-3) together with increased cleavage of its cellular substrate PARP (Fig. 1m). We observed that Fbxw7 overexpression markedly protected human HepG2 hepatocellular cells from Thapsigargin-induced ER stress and apoptosis, as well

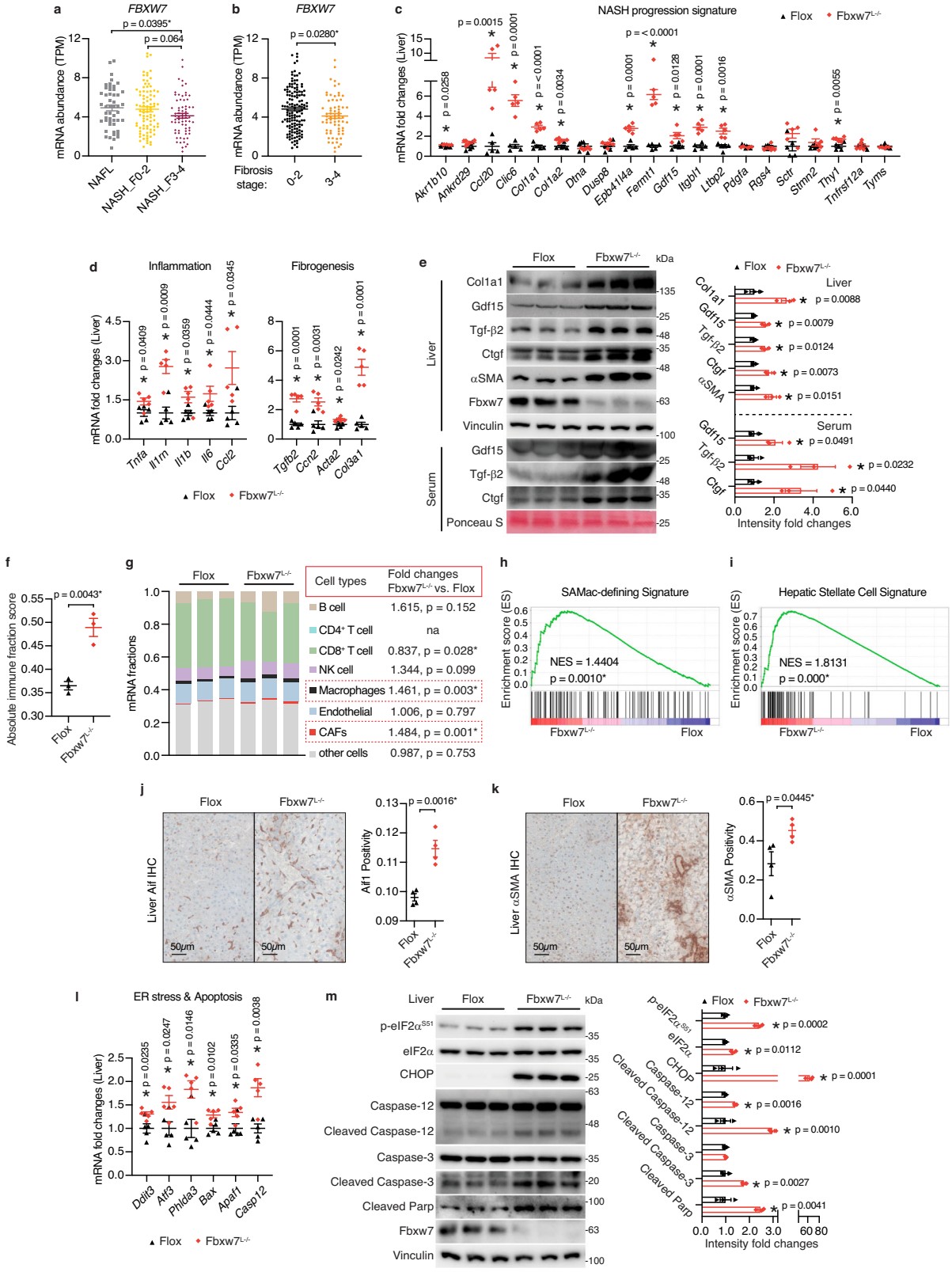

as resisted the induction of NASH progression signature genes (Supplementary Fig. 1o, p).

## Fat accumulation in Fbxw7-null livers is independent of hepatic lipogenesis confirmed by metabolome

Lipotoxicity is one of the potent inducers of persisting ER stress. *Fbxw7*[L−/−] mice displayed over two-fold hepatic TG content compared with control livers (Supplementary Fig. 2a), prompting us to investigate the source of the accreted liver fat caused by *Fbxw7* deletion. Gas chromatography–mass spectrometry (GC-MS) determination of hepatic profiles of free fatty acid (FFA) and total esterified lipids (TG, cholesterol ester, and phospholipid) revealed overall accretion of long-chain and very long-chain FAs in Fbxw7-null livers (Fig. 2a, b and Supplementary Data 1). Interestingly, we noted that Fbxw7 depletion

**Fig. 1 | FBXW7 protects against NASH. a** Analysis of hepatic *FBXW7* expression in patients across the severity stages of NAFLD progression. *n* = 50 for NAFL; *n* = 87 for NASH_F0-2; *n* = 68 for NASH_F3-4. **b** *FBXW7* mRNA levels in fibrotic liver biopsies from patients with NAFLD. *n* = 137 for Fibrosis stage 0-2; *n* = 68 for Fibrosis stage 3-4. Relative mRNA levels of NASH progression signature genes (**c**), inflammation and fibrogenesis genes (**d**) in *Flox* (control) and Fbxw7-null livers, *n* = 5. **e** Immunoblots and quantification of proteins related to NASH progression in livers and serum from *Flox* and *Fbxw7*$^{-/-}$ littermates. Each lane represents one mouse, *n* = 3. **f** Overall immune contents of control and Fbxw7-null livers, reflected by the absolute immune fraction scores, *n* = 3. **g** Stacked bar chart illustrating cell distributions of control and Fbxw7-null livers, *n* = 3. CAFs Cancer-associated fibroblasts, na non-applicable. GSEA of SAMac-defining signature (**h**) and hepatic stellate cell signature (**i**). Array genes were ordered from the highest in Fbxw7-null livers (left) to the highest in control livers (right). Locations of genes in each signature are indicated by the vertical black bars. NES normalized enrichment score. Representative images and quantification of Aif1 (**j**) and αSMA (**k**) immunohistochemistry (IHC) staining, *n* = 4. Scale bars, 50 μm. **l** mRNA expression of ER stress and apoptotic genes in control and Fbxw7-null livers, *n* = 5. **m** Immunoblots and quantification of proteins related to ER stress and apoptosis. Each lane represents one mouse, *n* = 3. Data are presented as means ± SEM (**a**–**f**, **j**–**m**). *$p < 0.05$, unpaired two-tailed Student's *t* test (**a**–**g**, **j**–**m**). Source data are provided as a Source Data file.

resulted in drastic accumulation of polyunsaturated fatty acid (PUFA) C18:2n6 (linoleic acid) and C18:3n3 (α-linolenic acid), the two essential unsaturated FAs that cannot be synthesized de novo and can only arise from diet or adipose tissue. n3 and n6 series of highly unsaturated FAs with 20- and 22-carbons, which are synthesized from dietary linoleic acid by the action of Δ5 and Δ6 desaturases (FADS1 and FADS2) were also elevated in Fbxw7-null livers without alterations in *Fads1/2* expression (Fig. 2b, Supplementary Fig. 2b, and Supplementary Data 1). *Fbxw7*$^{-/-}$ mice also exhibited a lower DNL index, calculated by the ratio of C16:0 to C18:2n6 (Fig. 2c), indicating that Fbxw7-depletion-induced accumulation of hepatic fat is unlikely to be derived from elevated endogenous biosynthesis. Lipogenesis is an adaptation to elevated glucose metabolism (Fig. 2d). GC-MS and liquid chromatography-mass spectrometry (LC-MS) measurement of liver metabolites demonstrated significantly reduced hepatic glucose and glycolytic/gluconeogenic intermediates (PEP, Alanine) in the absence of Fbxw7 (Fig. 2e and Supplementary Data 1). Additionally, *Fbxw7*$^{-/-}$ mice exhibited lower blood glucose and lactate levels (Fig. 2f) as well as remarkable defects in hepatic glycogen synthesis and glycogenolysis (Supplementary Fig. 2c, d), implying an inability to store and use glucose effectively. Consistently, ChREBP, a transcription factor (TF) activated by glucose influx to promote hepatic lipogenesis was greatly downregulated upon Fbxw7 depletion (Supplementary Fig. 2e). In accordance, lipogenic precursors acetyl-CoA and malonyl-CoA were significantly reduced in Fbxw7-null livers (Fig. 2e and Supplementary Data 1), partly contributed by drastic reduction of hepatic CoA abundance and downregulated CoA biosynthesis genes required for FA activation and acetyl-CoA production (Supplementary Fig. 2f, g). Other lipogenic intermediates such as citric acid, DHAP, glycerol, and glycerol-3P showed no difference between genotypes. Molecularly, glycolytic and gluconeogenic genes were generally downregulated in Fbxw7-null livers (*Gck, Ldha, Mpc1/2, Pdhb, G6pc, Pck1, Gpt1, Pcx*), while genes involved in FA synthesis (*Cs, Acly, Acaca/Acc1, Fasn*), elongation (*Elovl6*) and desaturation (*Scd1*) showed no differences between genotypes (Fig. 2g). Loss of Fbxw7 also repressed genes in glyceroneogenesis (*Gpd1, Gyk*) and subsequent TG synthesis (*Gpat4, Agpat2, Lipin1/2, Dgat1/2*) (Fig. 2g). These results were further confirmed at the protein level via immunoblots (Fig. 2h). Enzymatic assays revealed no changes in hepatic activity of acetyl-CoA carboxylase (ACC), the first and rate-limiting DNL enzyme that carboxylates acetyl-CoA to form malonyl-CoA, as well as comparable hepatic activity of fatty acid synthase (Fasn), a key enzyme in the synthesis of long-chain saturated FAs (Fig. 2i, j). Additionally, we examined hepatic lipogenic genes during refeeding, when the transcription and proteolytic processing of SREBP-1 are both active and observed negligible differences between genotypes (Supplementary Fig. 2h–j). These results further exclude the possibility that hepatic steatosis of *Fbxw7*$^{-/-}$ mice is caused by elevated lipogenesis.

The accumulation of essential unsaturated FAs in Fbxw7-null livers led us to investigate whether they were derived from elevated dietary uptake. *Fbxw7*$^{-/-}$ mice exhibited markedly lower (-34.4% of *Flox* control) circulating TG levels without changes in hepatic genes involved in chylomicron remnant uptake (*Ldlr, Lrp1*) (Fig. 2k and Supplementary Fig. 2k). Alternatively, *Apoc2* and *Apoa4*, Lpl coactivators that facilitate TG clearance by peripheral tissues, were significantly elevated in *Fbxw7*-null livers (Fig. 2l). Apoa4 simultaneously facilitates glucose uptake in both muscle and adipose tissue, storing external or de novo synthesized FAs as TGs in lipid droplets (LDs). Accordingly, epididymal white adipose tissue (eWAT) from *Fbxw7*$^{-/-}$ mice displayed elevated expression of genes and proteins involved in glucose uptake and lipogenesis (Fig. 2m, n) while showing no differences in weight compared to *Flox* controls (Supplementary Fig. 2l). Concordant with the fact that fat mass is kept by balanced synthesis and lipolysis, increased phosphorylation of hormone-sensitive lipase (Hsl), an indicator of activated lipolysis, was observed in eWAT from *Fbxw7*$^{-/-}$ mice despite comparable expression of lipolysis genes (Fig. 2m, n). Serum FFAs, which mainly originate from adipose tissue through lipolysis and contribute to approximately 60% of FAs used for hepatic TG synthesis, tended to be higher in *Fbxw7*$^{-/-}$ mice (Fig. 2o), consistent with their elevated liver FFA pool. *Fbxw7*$^{-/-}$ mice also displayed significantly reduced fed insulin levels (Fig. 2p), which might contribute to their increased lipolysis and adipose tissue loss. Collectively, our results reveal that enhanced liver TG and FA accrual in *Fbxw7*$^{-/-}$ mice occurs independently of a lipogenic program reflected at least in part by elevated lipolytic flux from adipose tissue.

## Loss of Fbxw7 impairs PPARα-dependent lipid oxidation
Hepatic fat homeostasis is maintained by an interplay between fat synthesis, uptake, oxidation, and export as very low-density lipoprotein (VLDL). To shed light on the major contributor to the fatty liver of *Fbxw7*$^{-/-}$ mice, we comprehensively analyzed genes participating in every aspect of fat metabolism according to the high-throughput transcriptome data (Fig. 3a). Liver RNA-seq analysis confirmed that genes in lipogenesis or chylomicron remnant uptake did not differ between genotypes while hepatic FA uptake genes showed a mixed pattern of changes (Fig. 3a and Supplementary Fig. 3a). On the other hand, *Fbxw7*$^{-/-}$ mice displayed impaired VLDL assembly and secretion (Fig. 3a and Supplementary Fig. 3b) in line with their significantly lower circulating TG levels (Fig. 2k). Most strikingly, genes regulating each step of fat degradation occurring in different cellular compartments were concurrently downregulated in Fbxw7-null livers, from initiation of hepatic TG hydrolysis in the cytosol or lysosomes to FA oxidation in mitochondria, peroxisomes, and microsomes (Fig. 3a). Consistently, GO biological process analysis revealed that catabolism of amino acids and FAs were the top two suppressed pathways in Fbxw7-null livers (Supplementary Fig. 3c). Moreover, mRNA and protein expression of AMPK were significantly downregulated in Fbxw7-null livers although its phosphorylation status did not differ (Supplementary Fig. 3d, e). For fat degradation to occur, TG must first be hydrolyzed into FAs via cytosolic or lysosomal lipases represented by adipose triglyceride lipase (Atgl, encoded by *Pnpla2*) and lysosomal acid lipase (Lal, encoded by *Lipa*). Mitochondrial and peroxisomal β-oxidation are the primary pathways for FA degradation. For complete FAO, FAs first need to be activated and transported into mitochondria by carnitine palmitoyl transferase (CPT)1, CPT2 and carnitine-acylcarnitine translocase (CACT, encoded by *Slc25a20*) followed by a variety of enzymatic

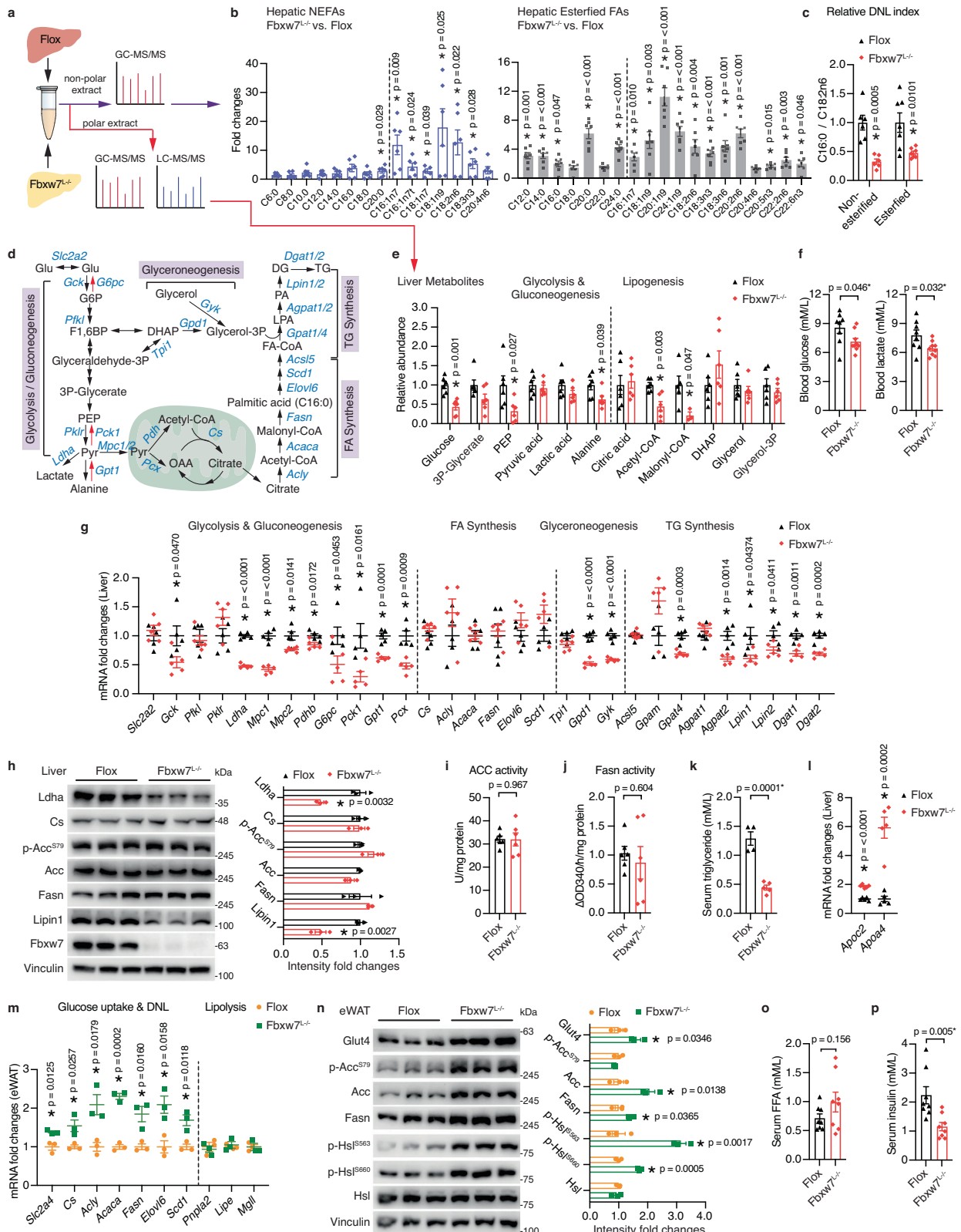

reactions that ultimately oxidize fatty acyl-CoAs to form acetyl-CoA (Fig. 3b). Fbxw7-null livers displayed drastically reduced levels of diverse fatty acyl-CoAs (Fig. 3c), which taken together with decreased acetyl-CoA and augmented FAs (Fig. 2b, e), indicate a diminished FAO capacity. Indeed, enzymatic assays confirmed strikingly attenuated FAO activity in Fbxw7-null livers (Fig. 3d). Concordant with RNA-seq analysis, RT-qPCR further validated that virtually all genes involved in

hepatic TG degradation were significantly downregulated upon Fbxw7 depletion (Fig. 3e). Noticeably, most of these genes are in the metabolic cascade of PPARα, a master regulator of hepatic lipid homeostasis[26] and an appealing target for anti-NASH clinical trials[39]. Indeed, 50% of PPARα targets[40] found within the mouse Hallmark FA metabolism gene signature (https://www.gsea-msigdb.org/gsea/) were significantly downregulated in Fbxw7-null livers (Supplementary

**Fig. 2 | Fat accumulation in Fbxw7-null livers is independent of hepatic lipogenesis. a** Schematic representation of *Fbxw7*[−/−] and *Flox* liver metabolomics study. See methods for further details. **b** Relative levels of free fatty acid (FFA) and esterified FA in *Fbxw7*[−/−] mice versus *Flox* mice. *n* = 6 for FFA and *n* = 7 for esterified FA unless otherwise indicated in Supplementary Data 1 due to missing values. **c** Relative de novo lipogenesis (DNL) index measured as the ratio of hepatic C16:0 content to C18:2n6 content (*n* = 6 for nonesterified, *n* = 7 for esterified). **d** Schematic illustration of intermediate metabolites and genes involved in glycolysis/gluconeogenesis, FA synthesis, glyceroneogenesis and triacylglyceride (TG) synthesis. Glu glucose, G6P glucose 6-phosphate, F1,6BP fructose 1,6-bisphosphate, PEP phosphoenolpyruvate, Pyr pyruvate, OAA oxaloacetate, DHAP dihydroxyacetone phosphate, LPA lysophosphatidic acid, PA phosphatidic acid, DG diacylglycerol. **e** Relative abundance of selected metabolites involved in glucose

metabolism and lipogenesis illustrated in Fig. 2d. *n* = 6 unless otherwise indicated in Supplementary Data 1 due to missing values. **f** Fed blood glucose and lactate levels of *Flox* (*n* = 8) and *Fbxw7*[−/−] (*n* = 9) mice. **g** Liver mRNA levels of genes illustrated in Fig. 2d in *Flox* and *Fbxw7*[−/−] mice, *n* = 5. **h** Immunoblots and quantification of proteins involved in glucose and fat metabolism. Each lane represents one mouse, *n* = 3. Hepatic ACC (**i**) and Fasn (**j**) activities in control and Fbxw7-null livers, *n* = 6. **k** Serum triglyceride concentrations of fed *Flox* (*n* = 4) and *Fbxw7*[−/−] (*n* = 5) mice. **l** Liver mRNA levels of *Apoc2* and *Apoa4*, *n* = 5. mRNA (**m**) and protein levels (**n**, each lane represents one mouse) of genes involved in glucose and fat metabolism in epididymal white adipose (eWAT) of *Flox* and *Fbxw7*[−/−] mice, *n* = 3. Serum FFA (**o**) and insulin (**p**) concentrations of fed *Flox* and *Fbxw7*[−/−] mice, *n* = 8. Data are presented as means ± SEM, *\*p* < 0.05, unpaired two-tailed Student's *t* test (**b**, **c**, **e–p**). Source data are provided as a Source Data file.

Data 2). Hepatic *Ppara* itself also showed concurrent reduction in *Fbxw7*[−/−] mice (Fig. 3f). Importantly, like *FBXW7*, *PPARA* expression declines as NASH progresses and is significantly reduced in liver fibrosis, cirrhosis, and HCC, with low *PPARA* expression associated with a poor survival rate among HCC patients (Fig. 3g, h and Supplementary Fig. 3f–h). Hepatic *PPARA* expression is positively correlated with *FBXW7* in patients with NAFL, NASH, liver fibrosis, and HCC (Fig. 3i, j and Supplementary Fig. 3i), implying their close association and pivotal contribution to liver health. Immunoblots further confirmed the drastic reduction of PPARα protein in Fbxw7-null livers together with significant decreases of multiple critical proteins regulating fat degradation (Fig. 3k). We further proved that impaired hepatic fat catabolism in *Fbxw7*[−/−] mice was directly caused by loss of Fbxw7 rather than long-term compensation, as acute Fbxw7 depletion via injection of *Flox* mice with adenovirus expressing control or CRE recombinase led to significant reductions in mRNA and protein levels of hepatic PPARα and FAO genes (Fig. 3l, m and Supplementary Fig. 3j). Furthermore, transient and stable deletion of FBXW7 in HepG2 cells both resulted in remarkable attenuation of PPARα mRNA and protein levels (Supplementary Fig. 3k–n). Consistently, transiently and stably overexpressing FBXW7 in HepG2 cells upregulated PPARα transcripts and protein abundance accompanied with augmented FAO gene expression, with the induction being more evident upon stable FBXW7 overexpression (Supplementary Fig. 3o–s). Interestingly, although FBXW7 stimulates PPARα in a cell-autonomous manner, immunoprecipitation (IP) experiments demonstrated no physiological interaction between endogenous Fbxw7 and PPARα in mouse liver (Supplementary Fig. 3t). This observation was further supported by IP studies performed in HepG2 cells treated with proteasome inhibitor MG132 as well as in 293T cells overexpressing FBXW7 and PPARα (Supplementary Fig. 3u, v), implying indirect regulation of PPARα by FBXW7. Together, these results suggest that the accumulated TG in Fbxw7-null livers is predominantly caused by their massive defects in fat clearance associated with attenuated PPARα expression.

### Fbxw7-null livers exhibit mitochondrial dysfunction and autophagy deficiency

We next sought to reveal the mechanisms underlying suppressed fat degradation in *Fbxw7*[−/−] mice. GO cellular component analysis demonstrated that downregulated DEGs identified in Fbxw7-null livers were most enriched in the mitochondrial matrix (Fig. 4a), in line with the fact that FAs are primarily catabolized in mitochondria. Mitochondrial DNA contents were significantly lower in Fbxw7-null livers partly caused by reduced *Cmpk2* (Fig. 4b and Supplementary Fig. 4a), which encodes a rate-limiting enzyme supplying deoxyribonucleotides for mtDNA synthesis. In addition to impaired mitochondrial biogenesis, Fbxw7-depleted hepatocytes displayed severe mitochondrial damage, as shown by elevated extracellular leakage of cytochrome c (Fig. 4c), a clinical marker of mitochondrial and cellular injury[41]. Cytochrome c release could be induced by excessive calcium fluxes to mitochondria from the ER and consequent opening of the

mitochondrial permeability transition pore. *Itpr3*, which encodes the main channel of ER calcium efflux, was significantly elevated in Fbxw7-null livers as well as in advanced NASH patients (Supplementary Fig. 4b, c). Damaged mitochondria are especially prone to inducing apoptosis via a caspase-9-dependent cascade initiated by cytochrome c leakage. Indeed, Fbxw7-null livers showed enhanced caspase-9 cleavage (Fig. 4d), which may also contribute to caspase-3 activation (Fig. 1m). We also found that Fbxw7-null livers displayed reduced expression of essential genes involved in multiple steps of autophagy, including initiation (*Ulk1*), vesicle nucleation (*Atg9a*), autophagosome membrane elongation (*Atg7, Atg5, Atg4a, Trp53inp2, Atg3*), maturation (*Gabarapl1/2*), substrate capture (*Sqstm1*), autophagosomes trafficking and fusion with lysosomes (*Uvrag, Stx17*), as well as lysosomal degradation (*Lamp1, Lamp2, Atp6v1b2, Ctsb*) (Fig. 4e). Autophagy deficiency in Fbxw7-null livers was further evidenced by reduced protein levels of essential autophagy components and markedly accumulated p62/Sqstm1 protein (Fig. 4f). Mitochondrial clearance can also be achieved through mitophagy, and genes encoding several mitophagy components inducing *Pink1, Prkn* and *Optn* were significantly downregulated in Fbxw7-null livers (Fig. 4g). Immunoblots further revealed accumulated hepatic Pink1 protein in Fbxw7-null livers, together with downregulated Parkin and mitophagy receptor proteins (Fig. 4h). Additionally, Fbxw7-null livers displayed inadequate mitochondrial remodeling, reflected by reduced fission and fusion proteins (Fis1, Drp1, Mff, Opa1), which are essential for dilution of mitochondrial damage, autophagy, and mito-biogenesis (Fig. 4i). Given that excessive lipid overload in turn represses the autophagy apparatus, we then tested whether the impaired autophagy in Fbxw7-null livers is a causative factor or a consequence of fat accumulation. We observed that acute Fbxw7 ablation significantly suppressed the mRNA and protein levels of key autophagy and mitophagy components together with accumulated p62 and Pink1 proteins (Fig. 4j, k), indicating autophagy attenuation and mitochondrial damage. In contrast, FBXW7 overexpression elevates autophagy genes in HepG2 cells (Supplementary Fig. 4d). Overall, Fbxw7-null livers manifest mitochondrial dysfunction and defective clearance of damaged mitochondria by autophagy, further attenuating fat catabolism and promoting apoptosis.

### Fbxw7[−/−] mice adapt poorly to starvation

Severe defects of *Fbxw7*[−/−] mice in mitochondrial fat catabolism and autophagy prompted us to investigate the role of Fbxw7 in energy metabolism. For maximal energy production, acetyl-CoA derived from glycolysis, FAO or amino acids metabolism enters the tricarboxylic acid (TCA) cycle followed by complete oxidation through oxidative phosphorylation (OXPHOS) (Fig. 5a). Gene expression and immunoblotting revealed that Fbxw7 depletion attenuated hepatic TCA cycle and OXPHOS components (Fig. 5b, c and Supplementary Fig. 5a) and elevated hepatic *Ucp2* expression (Supplementary Fig. 5b), resulting in significantly compromised ATP synthesis (Fig. 5d), thus increasing the susceptibility of *Fbxw7*[−/−] mice to conditions of acute energy demand.

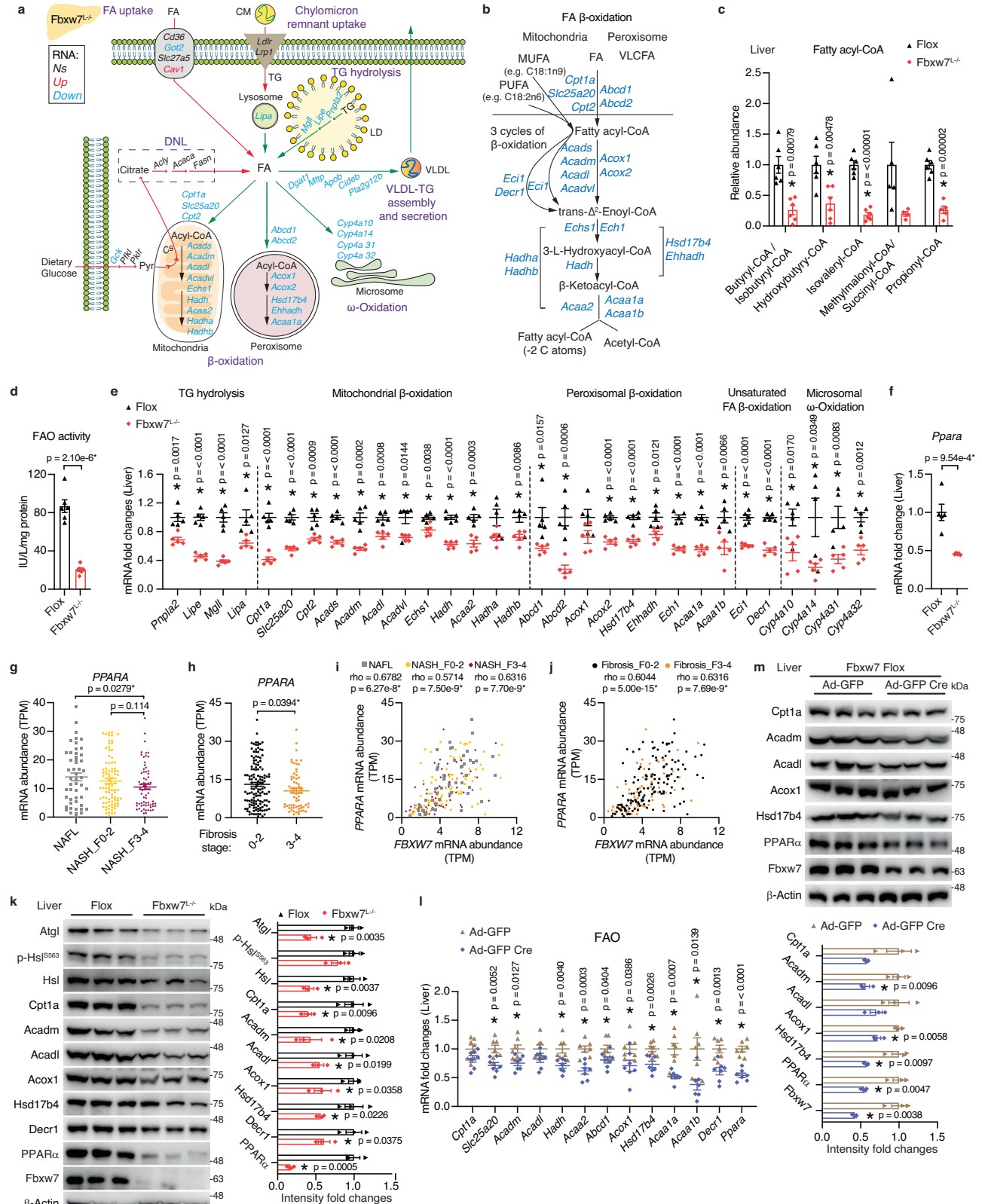

Indeed, prolonged food deprivation, which is known to shift energy fuel to fat as well as induce autophagy for nutrient reclamation, markedly aggravated the fatty liver phenotype of *Fbxw7⁻/⁻* mice as indicated by increased liver paleness with unchanged liver weights (Fig. 5e, f and Supplementary Fig. 5c). Fbxw7-null livers also exhibited diminished hepatic L-amino acid contents upon starvation (Fig. 5g), indicating impaired amino acid recycling, an initial role of autophagy.

Consistently, fasting-induced hepatic FAO and autophagy was blocked in *Fbxw7⁻/⁻* mice, as evidenced by attenuated Cpt1a induction, defective LC3B-II transformation from LC3B-I, as well as accumulated p62 and polyubiquitinated proteins (Fig. 5h and Supplementary Fig. 5d). *Fbxw7⁻/⁻* mice also experienced a steeper drop in blood glucose levels after food deprivation (Supplementary Fig. 5e), underlaid by defects in hepatic catabolism, which is known to fuel glucose production.

**Fig. 3 | Loss of Fbxw7 impairs PPARα-dependent lipid oxidation. a** Schematic diagram of genes regulating diverse aspects of hepatic fat metabolism. Significantly downregulated (blue), upregulated (red), and non-significant (black) genes from liver RNA-seq analysis of *Fbxw7*$^{L-/-}$ versus *Flox* mice ($n = 3$, $p < 0.05$, |FC| > 1.30). Red arrows, processes leading to accumulation of intrahepatic TG, green arrows, fat clearance processes. CM Chylomicron, LD Lipid Droplet. **b** Schematic illustration of intermediate metabolites and genes involved in each step of FA β-oxidation (FAO). VLCFA Very long chain FA, MUFA Monounsaturated FA, PUFA Polyunsaturated FA. **c** Relative abundance of fatty acyl-CoA in Fbxw7-null and control livers revealed by LC-MS. $n = 6$ unless otherwise indicated in Supplementary Data 1. **d** Hepatic FAO activities in control and Fbxw7-null livers, $n = 6$. **e** RT-qPCR analysis of genes in Fig. 3a, b regulating fat degradation in control and Fbxw7-null livers, $n = 5$. **f** Hepatic *Ppara* transcript levels, $n = 5$. **g** PPARA expression in patients across the severity

stages of NAFLD progression. $n = 50$ for NAFL; $n = 87$ for NASH_F0-2; $n = 68$ for NASH_F3-4. **h** PPARA mRNA levels in fibrotic liver biopsies from patients with NAFLD. $n = 137$ for Fibrosis stage 0-2; $n = 68$ for Fibrosis stage 3-4. Correlation analysis of FBXW7 and PPARA mRNA levels in human livers with NAFLD progression (**i**, see Fig. 3g for sample numbers) and human fibrotic liver biopsies (**j**, see Fig. 3h for sample numbers). **k** Immunoblots and quantification of proteins involved in hepatic fat degradation. Each lane represents one mouse, $n = 3$. Fbxw7 *Flox* mice were injected with adenovirus expressing GFP or Cre recombinase and livers were collected and subjected to RT-qPCR (**l**, $n = 7$) and immunoblot (**m**, $n = 3$) analysis of FAO genes. Each lane represents one mouse. Immunoblot quantification results are shown on the lower panel (**m**). Data are presented as means ± SEM, *$p < 0.05$, non-parametric Spearman's test (two-tailed) (**i, j**), unpaired two-tailed Student's *t* test (**c–h, k–m**). Source data are provided as a Source Data file.

Furthermore, nutrition deprivation elicited differences in body weight, with *Fbxw7*$^{L-/-}$ mice significantly losing more weight, caused by loss of muscle and fat mass (Fig. 5i). Severe muscle wasting in *Fbxw7*$^{L-/-}$ mice was accompanied by a prominent upregulation of genes involved in proteasome- and lysosome-mediated muscle atrophy alongside slightly increased FAO genes with comparable expression of glycogen metabolism genes (Fig. 5j, k). In line with overactivated muscle proteolysis, *Fbxw7*$^{L-/-}$ mice displayed resistance to fasting-induced decline in circulating amino acids (Fig. 5l). In addition to decreased adipose mass, *Fbxw7*$^{L-/-}$ mice possessed markedly smaller adipocytes and slightly higher serum FFAs upon fasting (Fig. 5m, n and Supplementary Fig. 5f). Adipose LD surface proteins, known to shield LDs from lipolysis, were significantly downregulated in fasted *Fbxw7*$^{L-/-}$ mice together with elevated Atgl protein and phospho-dependent activated Hsl (Fig. 5o, p), resulting in substantial adipose FA mobilization into Fbxw7-null livers upon fasting. Elevated autophagy and FAO were also observed and might have partly contributed to the loss of adipose mass in starved *Fbxw7*$^{L-/-}$ mice (Fig. 5o, p). Together, we demonstrated that *Fbxw7*$^{L-/-}$ mice harbor an increased vulnerability to energy deficiency associated with impaired TCA cycle and OXPHOS, resulting in elevated peripheral catabolism as a compensation.

### Hepatic Fbxw7 deficiency inhibits ketogenic diet-induced fat utilization

We further challenged the Fbxw7-dependent signaling system with an extremely high-fat (93.5%) and low-carbohydrate (1.8%) ketogenic (keto) diet, a chronic stimulus known to induce hepatic FAO and ketosis. Three weeks of keto diet feeding significantly induced weight loss, with *Fbxw7*$^{L-/-}$ mice losing twice more weight than the control mice (Fig. 6a). We next utilized indirect calorimetry to determine the impact of loss of hepatic Fbxw7 on whole-body energy homeostasis under a normal diet (ND) and keto diet. We observed no obvious differences in $O_2$ consumption, $CO_2$ production, energy expenditure (EE), respiratory exchange ratio (RER), or locomotor activity between genotypes during *ad libitum* ND feeding (Fig. 6b, c and Supplementary Fig. 6a–h), implying that hepatic defects in *Fbxw7*$^{L-/-}$ mice could be compensated when maintained under a normal energy state. Acute switching to a keto diet forced mice to rely primarily on fat as the energy fuel, reflected by dramatically reduced $CO_2$ production and consequent RER decline (Supplementary Fig. 6a, b, e, f). *Fbxw7*$^{L-/-}$ mice were less adept in shifting fuel sources to fat and evidently consumed less $O_2$ upon switching to a keto diet, underscored by their significantly blunted ability to activate hepatic FAO in comparison with their *Flox* littermate controls (Fig. 6b–e). Surprisingly, keto diet-fed *Fbxw7*$^{L-/-}$ mice displayed comparable daily calorie intake, reduced EE, and increased locomotor activity compared to ND-fed mice (Supplementary Fig. 6c, d, g–i). Keto diet-induced weight loss in *Fbxw7*$^{L-/-}$ mice was mainly derived from muscle and adipose tissue, accompanied by elevated circulating levels of amino acids and FFAs, the latter coupled to robustly increased adipose lipolysis in comparison with the *Flox* controls (Fig. 6f–j). A keto diet is known to stimulate adipose mobilization

to liver for oxidation with the generated acetyl-CoA either used in the TCA cycle for energy production or for the synthesis of ketone bodies (ketogenesis), which in turn curtails muscle proteolysis and adipose lipolysis by serving as an alternative energy fuel (Fig. 6k). A keto diet triggered liver fat accumulation, especially in *Fbxw7*$^{L-/-}$ mice, accompanied by downregulation of essential ketogenesis genes (*Acat1*, *Hmgcs2*, *Hmgcl*, *Bdh1*) (Fig. 6l, m, and Supplementary Fig. 6j). Hepatic ketogenesis intermediates Aceto-Acetyl-CoA and HMG-CoA showed a nonsignificant trend toward lower expression in Fbxw7-null livers (Supplementary Fig. 6k), likely because they are also involved in the mevalonate pathway. Markedly, *Fbxw7*$^{L-/-}$ mice displayed attenuated induction of blood ketone bodies under a keto diet, resulting in continuous extrahepatic catabolism, increased propensity to carbohydrate metabolism and severe hypoglycemia (Fig. 6n and Supplementary Fig. 6f, l). *Fbxw7*$^{L-/-}$ mice are thus intolerant to a keto diet, caused by insufficient hepatic respiration, ketogenesis, and unconstrained peripheral proteolysis and lipolysis.

### Metabolic stress aggravates NASH in the absence of Fbxw7

We observed that the NASH phenotype of *Fbxw7*$^{L-/-}$ mice was greatly exacerbated under the ketogenic states (fasting or feeding a keto diet), evidenced by elevated expression of NASH-related genes (Fig. 7a), as well as apparent hepatocyte ballooning and collagen deposition visualized via H&E and Masson's trichrome (MTC) staining, respectively (Fig. 7b). Energy restriction is known to inhibit mammalian target of rapamycin (mTOR). However, mTOR signaling was aberrantly activated in Fbxw7-null livers under the ketogenic states, reflected by increased phosphorylation of mTOR and its target p70S6K (Fig. 7c, d). Consequently, phosphorylation of Ulk1 was significantly elevated in Fbxw7-null livers despite reduced total protein levels (Fig. 7c, d), denoting exacerbated autophagy suppression. mTOR signaling is also activated by ER stress, triggering apoptosis via JNK[42], whose phosphorylation was upregulated in Fbxw7-null livers (Fig. 7c, d). By contrast, FBXW7 overexpression markedly mitigated ER stress-triggered activation of mTOR and JNK in HepG2 cells (Fig. 7e). Interestingly, both hepatic mTOR and JNK are known to suppress the PPARα-mediated FAO pathway[43,44]. These disrupted hepatic nutrient and energy-sensing cascades in *Fbxw7*$^{L-/-}$ mice upon metabolic stress greatly perturbed communications with peripheral tissues, wherein hepatic secreted factors act as signal messengers. To identify secreted factors responsible for reprogrammed systemic metabolism in *Fbxw7*$^{L-/-}$ mice under stress stimuli, we employed an unbiased bioinformatic approach by overlapping the liver-derived secretome with hepatic DEGs induced by Fbxw7 depletion, fasting, or a keto diet (Fig. 7f). This approach yielded 10 specific factors known to have biological functions in lipoprotein metabolism (*Apoa4*), immunity and inflammation (*Apcs*, *Csf1r*, *Il1rn*), fibrogenesis (*Col14a1*), lipid droplet protein (*Hsd17b11*), iron homeostasis (*Bdh2*), complement activation (*C1qb*), blood coagulation (*F11*) and insulin-like growth factor-binding protein (*Igfbp1*), with *Apoa4*, *Apcs* and *Igfbp1* being the top three ranked hits based on

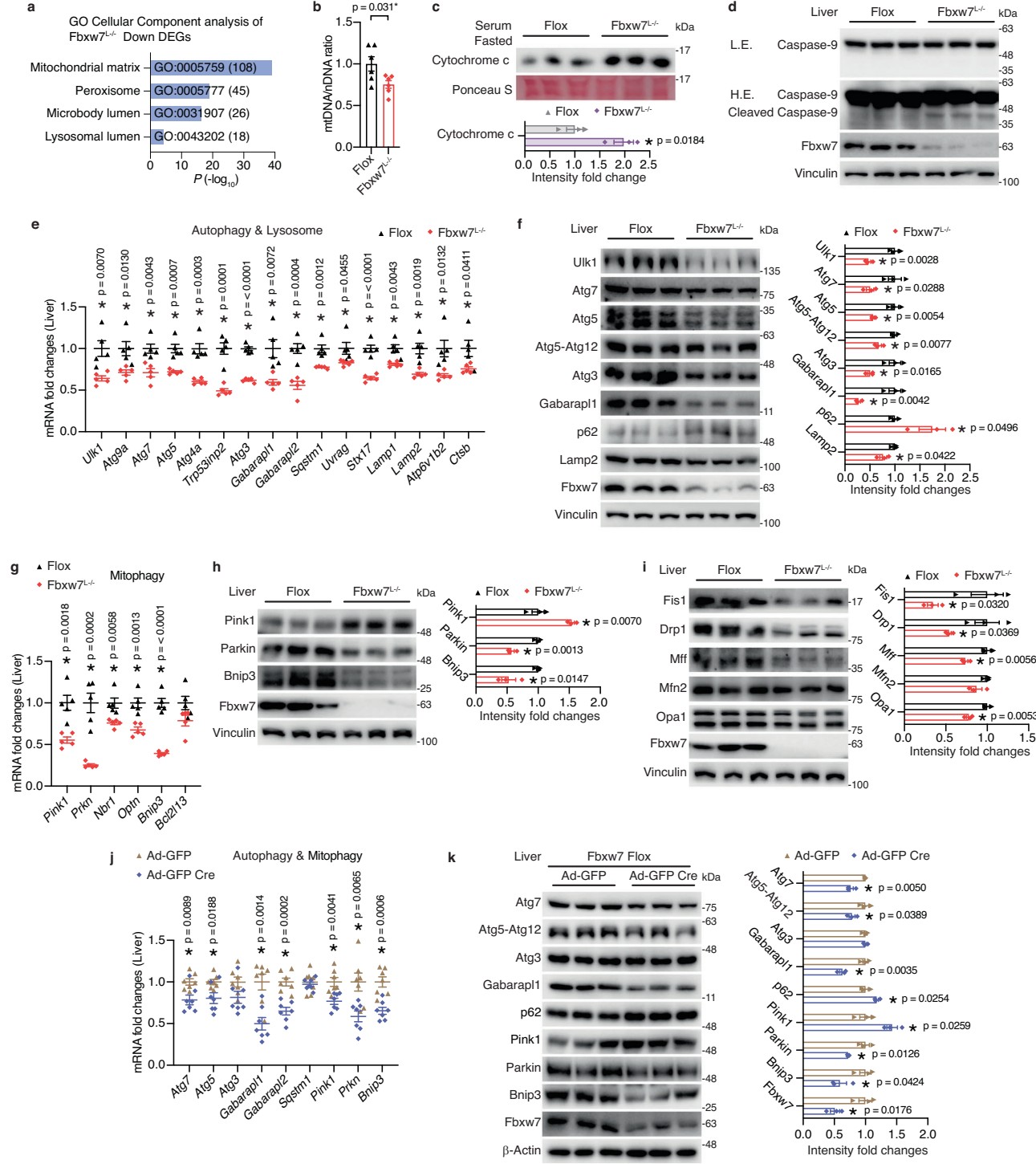

**Fig. 4 | Fbxw7-null livers exhibit mitochondrial dysfunction and autophagy deficiency. a** GO cellular component analysis of downregulated DEGs ($p < 0.05$, |FC| > 1.30) in Fbxw7-null livers. Top enriched terms are shown. The number of genes in each category are shown in parentheses. **b** Relative amounts of mitochondrial DNA (mtDNA), normalized to nuclear DNA (nDNA) in control and Fbxw7-null livers ($n = 6$). **c** Immunoblots and quantification of serum Cytochrome c levels of *Flox* and *Fbxw7*$^{L-/-}$ littermates. Each lane represents one mouse, $n = 3$. **d** Immunoblots and quantification of hepatic Caspase-9 and cleaved Caspase-9. Each lane represents one mouse, $n = 3$. L.E. Low Exposure, H.E. High Exposure. **e** mRNA expression of autophagy and lysosome-related genes in control and Fbxw7-null livers, $n = 5$. **f** Immunoblots and quantification of key autophagy

components. Each lane represents one mouse, $n = 3$. **g** mRNA expression of mitophagy genes in control and Fbxw7-null livers, $n = 5$. Immunoblots and quantification of key mitophagy components (**h**) and proteins mediating mitochondrial fission and fusion (**i**). Each lane represents one mouse, $n = 3$. **j, k** Fbxw7 *Flox* mice were injected with adenovirus expressing GFP or Cre recombinase and livers were collected and subjected to RT-qPCR (**j**, $n = 7$) and immunoblot (**k**, $n = 3$) analysis of autophagy and mitophagy genes. Each lane represents one mouse. Immunoblot quantification results are shown on the right panel (**k**). Data are presented as means ± SEM, *$p < 0.05$, unpaired two-tailed Student's *t* test (**b**, **c**, **e–k**). Source data are provided as a Source Data file.

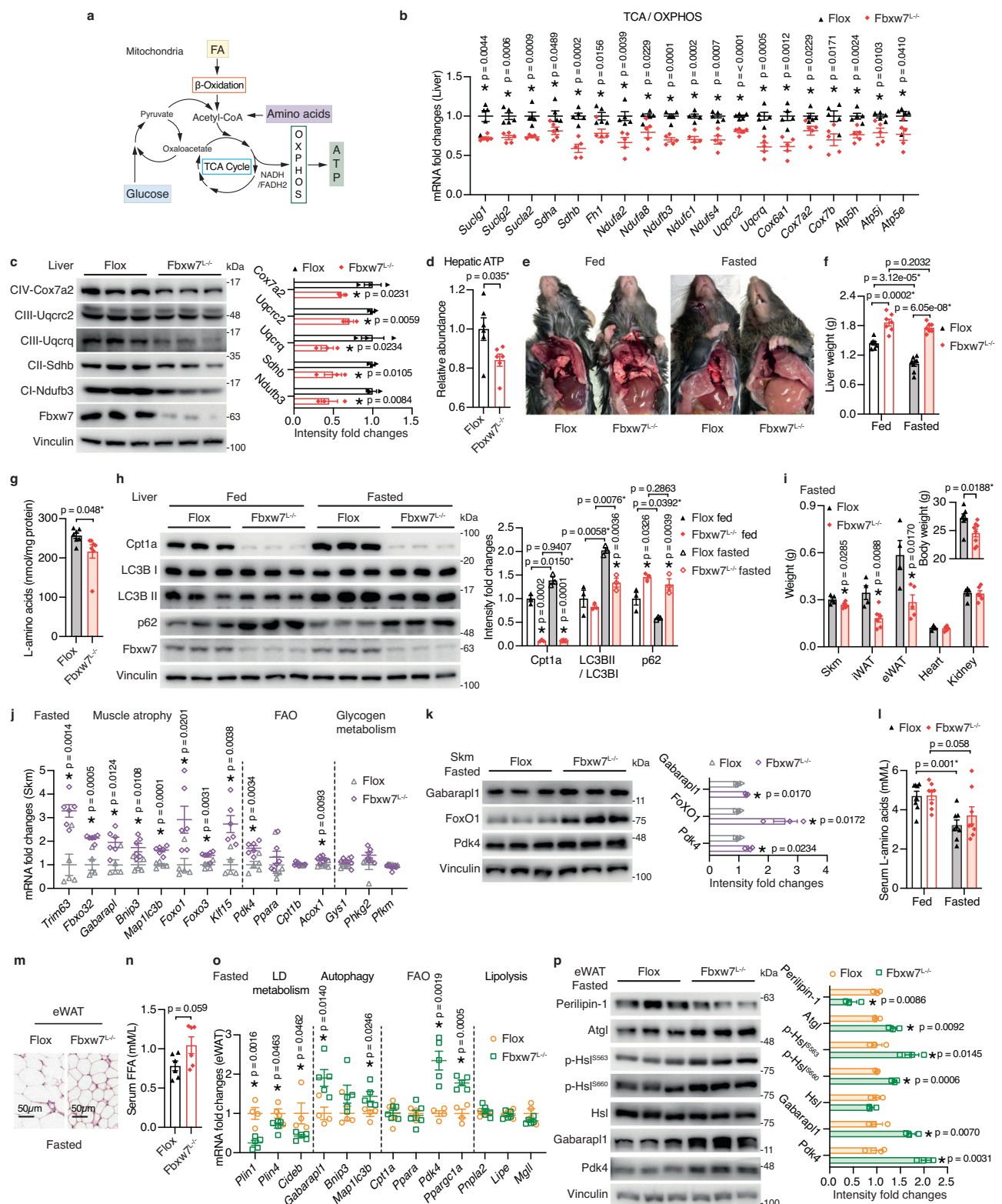

combined analysis of the four transcriptome datasets (Fig. 7f, g). Intriguingly, in addition to promoting peripheral uptake of TG and glucose, serum Apoa4 is significantly elevated in patients with early-stage liver fibrosis, promoting TGFβ-induced hepatic stellate cell activation and fibrogenesis, and has thus been proposed to serve as a biomarker for diagnosing hepatic fibrosis[45]. In contrast, serum amyloid P component (SAP/Apcs) decreases in plasma of human NASH patients, inhibiting fibrocyte differentiation[46]. Igfbp1, a hepatokine

implicated in both glucose and lipid homeostasis, has been reported to trigger adipose lipolysis upon a hepatic FAO defect[47]. *Apoa4* and *Igfbp1* transcripts were excessively elevated in Fbxw7-null livers while *Apcs* was transcriptionally repressed under the ketogenic states (Fig. 7h, i). Serum Apoa4, Apcs and Igfbp1 contents paralleled their hepatic mRNA expressions, showing slight discrepancies with liver protein levels, likely caused by secretion and export into the circulation (Fig. 7j, k).

**Fig. 5 | *Fbxw7*$^{-/-}$ mice adapt poorly to starvation. a** Illustration of hepatic energy metabolism. **b** mRNA expression of TCA cycle and OXPHOS genes in control and Fbxw7-null livers, *n* = 5. **c** Immunoblots and quantification of OXPHOS subunits. Each lane represents one mouse, *n* = 3. **d** Relative hepatic ATP abundance, *n* = 6. Gross liver morphology (**e**) and liver weights (**f**) of fed and fasted *Flox* and *Fbxw7*$^{-/-}$ mice. *n* = 7 for fed, *n* = 8 for fasted *Flox* and *n* = 7 for fasted *Fbxw7*$^{-/-}$ mice. **g** Liver L-amino acid contents of fasted *Flox* and *Fbxw7*$^{-/-}$ mice, *n* = 7. **h** Immunoblots and quantification of proteins involved in hepatic fat degradation and autophagy from *Flox* and *Fbxw7*$^{-/-}$ mice in the fed and fasted states. Each lane represents one mouse, *n* = 3. **i** Body weight and tissue weights of fasted *Flox* and *Fbxw7*$^{-/-}$ mice. See Source data file for sample numbers. mRNA (**j**, *n* = 5 for *Flox* and *n* = 6 for *Fbxw7*$^{-/-}$ mice) and protein levels (**k**, each lane represents sample from one mouse, *n* = 3) of the indicated genes in Skm of fasted *Flox* and *Fbxw7*$^{-/-}$ mice. Immunoblot quantification shown on the right panel (**k**). **l** Serum L-amino acids concentrations of *Flox* and *Fbxw7*$^{-/-}$ mice in the fed and fasted states, *n* = 8. **m** Representative hematoxylin and eosin (H&E)-stained eWAT from fasted *Flox* and *Fbxw7*$^{-/-}$ mice. Scale bars, 50 μm. See Supplementary Fig. 5f for quantification of eWAT size. **n** Serum FFA concentrations, *n* = 6. mRNA (**o**, *n* = 4 for *Flox* mice and *n* = 5 for *Fbxw7*$^{-/-}$ mice) and protein levels (**p**, each lane represents sample from one mouse, *n* = 3) of the indicated genes in eWAT from fasted *Flox* and *Fbxw7*$^{-/-}$ mice. Immunoblot quantification shown on the right panel (**p**). Data are presented as means ± SEM, *\*p* < 0.05, unpaired two-tailed Student's *t* test (**b–d**, **f–l**, **n–p**). Source data are provided as a Source Data file.

## Inhibition of Fbxw7 substrate ERRα alleviates NASH

Transcriptional regulation plays an indispensable role in NAFLD development as well as in controlling metabolic flexibility[23,48]. Given that the majority of Fbxw7 substrates are transcription factors and cofactors, we utilized the TRRUST database to identify transcriptional regulators downstream of Fbxw7 signaling. Previously characterized Fbxw7 substrates Junb, ERRα and Myc were predicted as top TFs regulating hepatic genes altered upon Fbxw7 depletion (Fig. 8a). Further investigation into the transcriptional regulation of nutrient-sensitive genes in Fbxw7-null livers predicted the nuclear receptors PPARα and ERRα as the top two TFs regulating genes co-affected by Fbxw7 and fasting (Fig. 8a). ERRα controls diverse aspects of hepatic metabolism[27] but its link to NASH development has not been explored. We first observed in a human NAFL/NASH cohort[35] that *ESRRA* expression positively correlates with NASH-related genes, especially during the NASH F0-2 stage (Fig. 8b). In agreement, these genes were generally upregulated in livers from HFD-fed ERRα[3SA] mice (Fig. 8c), a recently developed mouse model engineered to express an ERRα mutant protein which is no longer under the ubiquitylation-dependent control of Fbxw7, thus exhibiting ERRα overactivation[15]. Upregulation of these genes could be reversed by treatment with the ERRα inverse agonist C29[49] (Fig. 8c). In addition, global ERRα deficiency protected mice from HFD-induced elevation of key NASH-related genes as well as activation of proteins involved in ER stress and apoptosis (Fig. 8c and Supplementary Fig. 7a). RNA-seq followed by GSEA analysis revealed that loss of ERRα protected mice from HFD-induced enrichment of inflammation and apoptosis hallmark genes, as well as SAMac, hepatic stellate cell, and HCC-related progenitor cell signature genes (Fig. 8d, e and Supplementary Fig. 7b–d). GO cellular component analysis further revealed "Collagen-containing extracellular matrix" and "Endoplasmic reticulum lumen" as the top two terms enriched by DEGs identified in livers of HFD-fed ERRα KO mice, mostly downregulated in the absence of ERRα (Fig. 8f). Comparison of DEGs identified in livers from *Fbxw7*$^{-/-}$ mice and HFD-fed ERRα-null mice revealed 759 common genes, about half of which were downregulated in ERRα KO mice while simultaneously upregulated in *Fbxw7*$^{-/-}$ mice and were highly enriched in "Collagen-containing extracellular matrix" (Fig. 8f and Supplementary Fig. 7e). These contrasts in gene expression indicate that Fbxw7-mediated ERRα suppression plays a crucial role in NASH progression. To determine whether transcriptional dysregulation in Fbxw7-null livers associates with ERRα DNA binding, we performed ERRα ChIP followed by high-throughput sequencing (ChIP-seq) using chromatin from *Flox* control and Fbxw7-null livers in the fasted state (Supplementary Data 3 and 4). In accordance with elevated levels of ERRα protein, ChIP-seq analysis revealed a 2.6-fold increase in the number of ERRα-bound peaks within ±20 kb of gene transcription start sites (TSSs) and an over 5-fold increase within promoter regions in Fbxw7-null livers (Supplementary Fig. 7f–h). Moreover, we observed that 82% of DEGs (2834 of 3470) presented in Fbxw7-null livers were differentially bound by ERRα in comparison with control livers (FC ≥| 1.5|), including genes involved in FAO (*Cpt1a*), autophagy (*Bnip3, Atg7*), lipoprotein metabolism (*Apoa4*), inflammation (*Il1rn*) and NASH

progression (*Col1a1, Fermt1, Epb41l4a*), suggesting their reliance on an ERRα-dependent transcriptional network (Fig. 8g). Indeed, we observed via RNA-seq that loss of ERRα greatly prevented HFD-induced elevation of *Apoa4, Il1rn*, and *Col1a1* in the liver (Supplementary Fig. 7i). To examine whether ERRα is a main metabolic effector of NASH development, we injected *Fbxw7*$^{-/-}$ mice with C29, which resulted in significantly reduced hepatic neutral fat and TG contents upon clearance of excessive ERRα protein (Fig. 8h, i and Supplementary Fig. 7j). GC-MS analysis further revealed that administration of C29 tended to reduce hepatic FFAs and significantly promoted the clearance of hepatic long-chain esterified FAs including C20:0 (−55%), C18:3n3 (−50%), C18:3n6 (−52%), and C20:5n3 (−42%) (Supplementary Fig. 7k and Supplementary Data 5). ERRα is known to regulate FAO genes, which are generally repressed by ERRα in the liver[27]. Massive impairment of FAO in Fbxw7-null livers might be driven by competition for DNA binding between redundant ERRα and drastically reduced PPARα protein. To investigate this possibility, we compared our ERRα ChIP-seq data with an available liver PPARα ChIP-seq dataset (GSM1514930)[40] re-analyzed from raw fastq sequencing files using the same pipeline for ERRα. Noteworthy, the selected PPARα ChIP-seq dataset was performed on livers of mice treated with PPARα agonist GW7647, thus ensuring a strong catalog of PPARα-targeted genes. Examination of ERRα occupancy of PPARα target genes revealed a 64% overlap in *Flox* livers, with this percentage rising to 90% in Fbxw7-null livers (Fig. 8j). The augmentation in PPARα/ERRα co-targeted genes owing to Fbxw7 loss was largely ascribed to ERRα enrichment at established PPARα binding sites (Fig. 8j), signifying a context-dependent recruitment of ERRα to PPARα sites. Consensus ERR (ERRE) and PPAR (PPRE) response elements were identified in 38% of overlapping peaks (1552 of 4056), the majority (74%) constituting a hybrid ERRE/PPRE element predominantly manifesting as the ERRα motif CAAAGGTCA embedded within the PPARα DR1 motif AGGTCAAAGGTCA, reinforcing the potential competition between ERRα and PPARα (Fig. 8j and Supplementary Fig. 7l). A subset of Fbxw7-null liver DEGs marked by co-targeted loci harboring consensus ERRE and PPRE elements were found highly enriched in FAO (Supplementary Fig. 7m). ChIP-qPCR analyses performed on the same chromatin pools used for ERRα ChIP-seq study confirmed increased ERRα occupancy at several examined genes in Fbxw7-null versus *Flox* liver with a simultaneous decline in PPARα binding (Supplementary Fig. 7n). Together, these findings support that competition for occupancy between ERRα and PPARα indeed exists whereby attenuated PPARα activity in NASH would favor binding of ERRα to a specific set of genes.

The downregulation of PPARα transcription in hepatocytes lacking Fbxw7 (Fig. 3f, k–m and Supplementary Fig. 3k–n) together with increased ERRα recruitment to the PPARα locus in *Fbxw7*$^{-/-}$ mice (Supplementary Fig. 8a), suggests that Fbxw7-mediated regulation of PPARα is ERRα-dependent. In line with this notion, genetic and pharmacological inhibition of ERRα in HepG2 cells abrogated the observed reduction in PPARα mRNA and protein expression upon FBXW7 deletion (Supplementary Fig. 8b–f). Accordingly, suppressed expression of mRNA and proteins of essential genes involved in

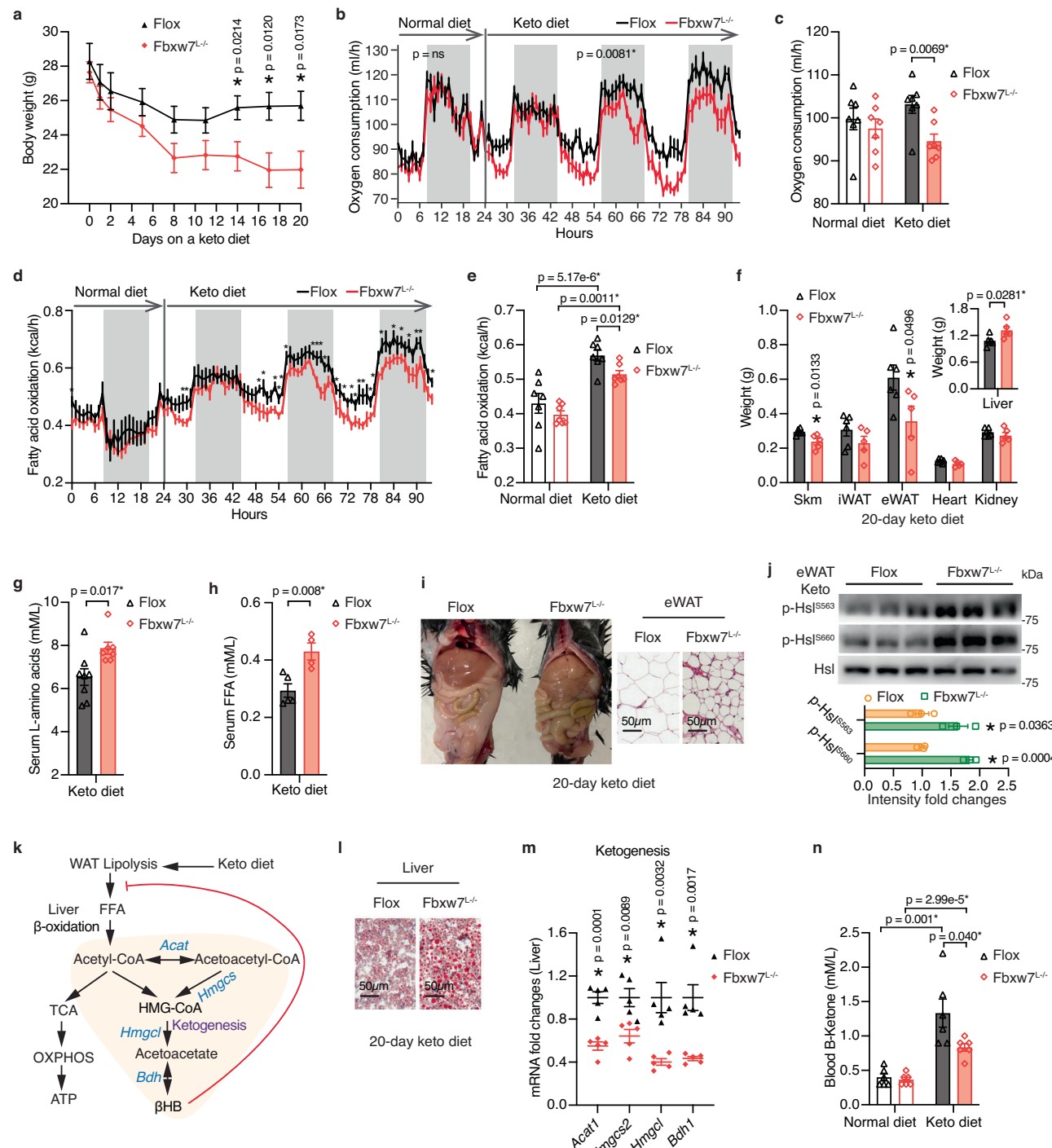

**Fig. 6 | Hepatic Fbxw7 deficiency inhibits ketogenic diet-induced fat utilization.**
**a** Body weight of *Flox* (*n* = 8) and *Fbxw7*[L−/−] (*n* = 7) mice during a 20-day keto diet. Hourly plot (**b**) and overall average (**c**) of whole-body oxygen consumption of *Flox* and *Fbxw7*[L−/−] mice switching from a normal diet to keto diet, *n* = 7. Hourly plot (**d**) and overall average (**e**) of fatty acid oxidation rate of *Flox* and *Fbxw7*[L−/−] mice switching from a normal diet to keto diet, *n* = 7. **f** Tissue weights of *Flox* and *Fbxw7*[L−/−] mice post a 20-day keto diet, *n* = 6 for *Flox* mice and *n* = 5 for *Fbxw7*[L−/−] mice. Concentrations of serum L-amino acids (**g**, *n* = 8 for *Flox* and *n* = 7 for *Fbxw7*[L−/−] mice) and FFA (**h**, *n* = 5 for *Flox* and *n* = 4 for *Fbxw7*[L−/−] mice) of mice post a 20-day keto diet. **i** Gross morphology (left) and representative eWAT H&E staining

(right) from *Flox* and *Fbxw7*[−/−] mice post a 20-day keto diet. Scale bars, 50 μm.
**j** Immunoblots and quantification of Hsl phosphorylation levels post a 20-day keto diet. Each lane represents eWAT from one mouse, *n* = 3. **k** Schematic illustration of intermediate metabolites and genes involved in ketogenesis. **l** Representative liver H&E staining of *Flox* and *Fbxw7*[−/−] mice post a 20-day keto diet. Scale bars, 50 μm. **m** RT-qPCR of hepatic levels of ketogenesis genes, *n* = 5. **n** Blood B-Ketone levels in *Flox* and *Fbxw7*[L−/−] mice before and after 16 days of keto diet, *n* = 6. Data are represented as means ± SEM (**a–h**, **j**, **m**, **n**). *p < 0.05, unpaired two-tailed Student's *t* test (**a**, **c–h**, **j**, **m**, **n**) and one-way ANCOVA using body weight as the covariate (**b**). Source data are provided as a Source Data file.

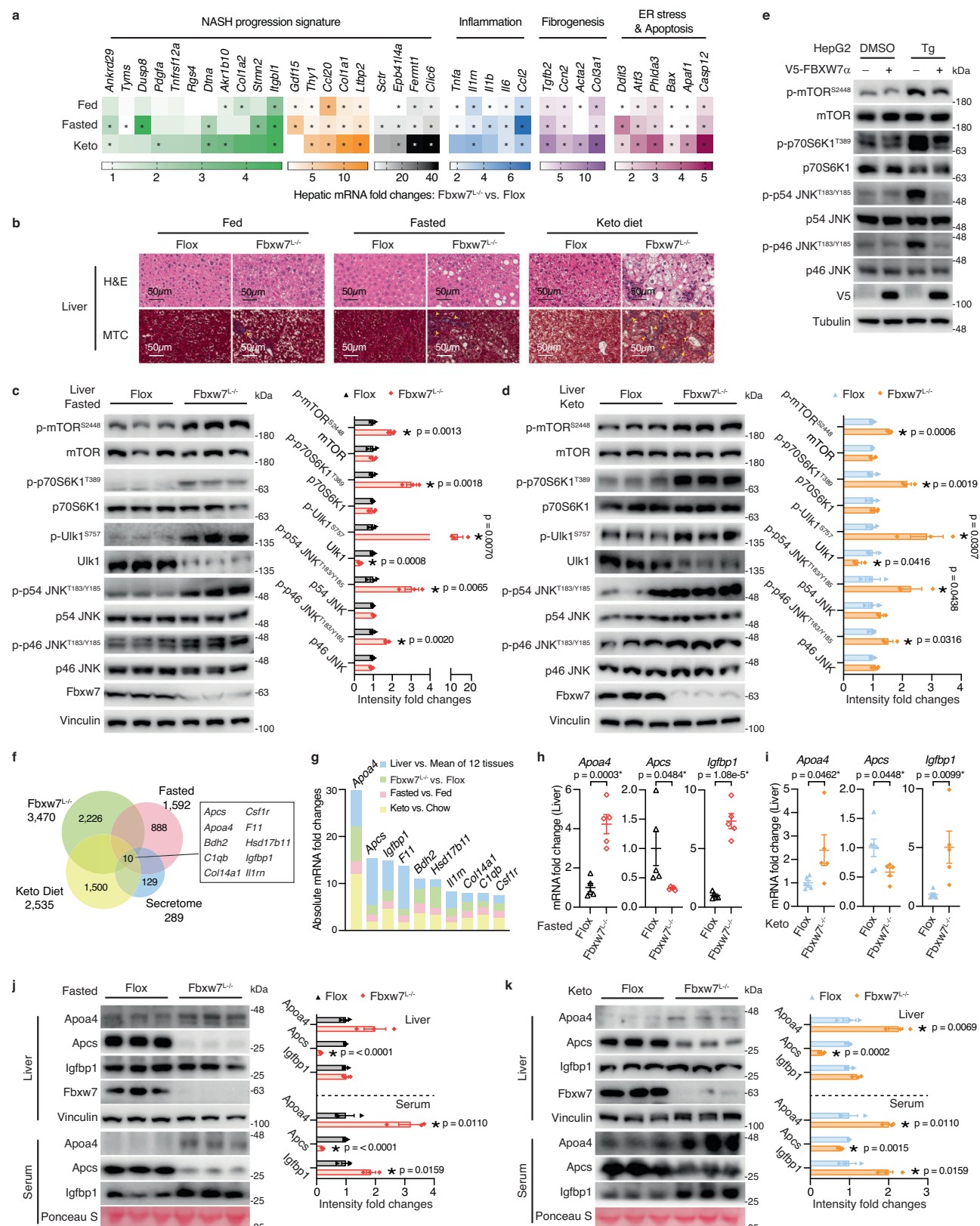

hepatic FAO were partially restored by C29 in *Fbxw7*[L−/−] mice, in line with augmented hepatic FAO activity (Fig. 8k and Supplementary Fig. 8g, h).

Targeting the expression of Fbxw7 itself might serve as an alternative therapy. ERRα is very likely to transcriptionally regulate Fbxw7 to control its own stability. Indeed, mouse liver ChIP-seq[25] and ChIP-

qPCR revealed ERRα recruitment to regulatory regions of the *Fbxw7* gene containing the consensus ERRE, and these DNA binding events could be abrogated by C29 (Supplementary Fig. 9a, b). Thus, ERRα depletion as well as C29 treatment elevated the mRNA and protein levels of Fbxw7 (Supplementary Fig. 9c–f), further indicating the benefits of ERRα inhibition for NASH treatment.

**Fig. 7 | Metabolic stress aggravates NASH in the absence of Fbxw7. a** Heatmap showing mRNA fold changes of NASH-related genes in the fed, fasted and keto diet nutritional regimens, assessed by RT-qPCR, *n* = 5-6 (see Source data file for specific sample numbers). **b** Representative images of H&E and Masson's trichrome (MTC) stained liver tissue sections from *Flox* and *Fbxw7*$^{L-/-}$ mice under different nutritional regimens. Arrowhead, fibrotic area (blue). Images are representative of *n* = 3 mice per group. Immunoblots and quantification of the indicated proteins from *Flox* and *Fbxw7*$^{L-/-}$ mice upon fasting (**c**) or post a 20-day keto diet (**d**). Each lane represents one mouse, *n* = 3. **e** HepG2 cells stably expressing empty vector or FBXW7 were treated with 500 nM Thapsigargin (Tg) or DMSO for 24 h followed by immunoblot examination. Immunoblots are from one experiment representative of 3

independent experiments with similar results. **f** Overlap of liver-enriched secreted factors with hepatic DEGs (*p* < 0.05, |FC|>1.30) induced by Fbxw7 deficiency, by fasting, or by a keto diet. **g** Ranking of the secreted factors identified in Fig. 7f by the sum of absolute mRNA fold changes of the four transcriptome datasets. **h, i** Hepatic mRNA expression of *Apoa4*, *Apcs* and *Igfbp1* from *Flox* and *Fbxw7*$^{L-/-}$ mice upon fasting (**h**, *n* = 5) or post a 20-day keto diet (**i**, *n* = 6 for *Flox* and *n* = 5 for *Fbxw7*$^{L-/-}$ mice). Immunoblots and quantification of serum and liver levels of the hepatic secreted factors from *Flox* and *Fbxw7*$^{L-/-}$ mice upon fasting (**j**) or post a 20-day keto diet (**k**). Each lane represents one mouse, *n* = 3. Data are represented as means ± SEM (**c**, **d**, **h**–**k**). *$p$ < 0.05, unpaired two-tailed Student's *t* test (**a**, **c**, **d**, **h**–**k**). Source data are provided as a Source Data file including Fig. 7a *p*-values.

## Discussion

Our study provides a comprehensive understanding of the metabolic abnormalities, from cellular processes and pathways to communication between different cell types and organ systems, that lead to primary lipid accretion and NASH pathogenesis in hepatocyte-specific *Fbxw7*-null mice, a genetic NASH mouse model. Using extensive omics approaches, including transcriptomics, bioinformatics, metabolomics, and genome-wide DNA binding studies, we revealed the E3 ligase Fbxw7 as a key regulator fine-tuning the transcriptional activity of the nutrient-sensing nuclear receptors PPARα and ERRα to stimulate hepatic catabolism and mitigate cellular stress (Fig. 9). Attenuated *FBXW7* and *PPARA* in advanced NASH patients along with uncontrolled ERRα activity synergistically create a vicious cycle in part through lipotoxicity and exacerbate the imbalance of hepatic and systemic homeostasis especially upon metabolic stress, underlying NASH progression. This work thus supports the notion that breaking this vicious cycle, either by increasing the expression of Fbxw7 or inhibiting ERRα, alleviates NASH.

Dysregulated autocrine and paracrine signals are known to drive NASH pathogenesis via a complex interplay between multiple cell lineages within the microenvironment[5,6]. For instance, macrophage contributions to liver inflammation and fibrosis have been characterized extensively[50,51]. A comprehensive study at the single-cell level has characterized a subpopulation of SAMacs that expand in NAFLD and promote liver fibrosis via regulating the activation and proliferation of hepatic stellate cells[36], which are believed to be the primary mediator of liver fibrogenesis upon transdifferentiation to myofibroblasts[50]. How hepatocytes connect with other intrahepatic cells to regulate NASH progression remains unknown. However, our data reveal that *Fbxw7*-depletion in hepatocytes, via stimulating macrophage-mediated production and secretion of inflammatory mediators, activates hepatic stellate cells toward fibrogenesis and contributes to NASH progression, implying that Fbxw7 integrates the physiological and metabolic activities of different cell types in the liver.

We observed multiple tightly connected cellular stress events in Fbxw7-null livers that contributed to NASH progression, including ER stress, mitochondrial dysfunction, autophagy suppression and activation of the apoptotic programme. The rate of FA disposal via mitochondrial β-oxidation was reported to be upregulated in NAFL patients as an adaptation to increased liver fat content[52]. However, this adaptation is lost in later stage-NASH patients, due to mitochondrial dysfunction[53]. Moreover, inefficient autophagy compromises mitochondrial turnover in Fbxw7-null livers and exacerbates their energetic imbalance. As a cytoprotective process, autophagy is closely associated with liver inflammation and fibrosis via setting a higher threshold for the induction of apoptosis through multiple mechanisms such as eliminating damaged mitochondria, recycling unwanted proteins, and maintaining bioenergetic homeostasis[54]. Our study suggests activating Fbxw7 as an anti-NASH therapy to break this vicious cycle by mitigating cellular stress and inhibiting apoptosis via direct control of the expression of its transcription factor targets.

We uncovered declined *FBXW7* and *PPARA* as features of advanced NASH and revealed significant correlations between *FBXW7*

and *PPARA* in multiple human liver diseases. It is well-known that hepatic fat catabolism strongly relies on PPARα-mediated transcriptional control. Accordingly, we observed that loss of Fbxw7 in hepatocytes downregulates the entire fat degradation network without affecting the lipogenic program, which is pivotal for NAFLD initiation but is unlikely to be the main contributor to NASH progression, suggesting that the accumulated TG in Fbxw7-null livers is predominantly caused by their massive defects in fat clearance associated with attenuated PPARα expression. Consequently, *Fbxw7*$^{L-/-}$ mice displayed severe fatty liver upon starvation and inadequate ketogenic response when feeding a keto diet. Interestingly, we found that even when deletion of *Fbxw7* is restricted to the liver, peripheral tissues such as eWAT and muscle were also affected under the ketogenic states. This intercommunication between metabolic organs is partly driven by deregulated liver-secreted macromolecules and proteins connecting liver and peripheral tissues. In response to fasting, the liver provides fuel sources for peripheral tissues via increasing hepatic glucose production through gluconeogenesis (GNG) as well as supplying ketones and lipids through ketogenesis and VLDL secretion. Glucose and fat metabolism are mutually influenced. Hepatic FAO is critical for GNG and ketogenesis, which are essential for systemic adaptations under metabolic stress. Defective FAO in *Fbxw7*$^{L-/-}$ mice resulted in repressed GNG as acetyl-CoA obtained via FAO is a potent GNG activator[55]. Interestingly, gluconeogenic enzymes seem to have a protective function against steatosis and NASH, since loss of gluconeogenic enzymes and transcription factors often result in NAFLD[56]. Given that circulating levels of ketone bodies inhibit adipose lipolysis, deficient ketogenesis further exacerbated imbalanced peripheral lipid mobilization and hepatic oxidation in *Fbxw7*$^{L-/-}$ mice accompanied by hypoglycemia. Liver also plays an important role in lipoprotein metabolism, secreting VLDLs to transport TGs to peripheral tissues[57]. Reduced VLDL assembly and secretion in Fbxw7-null livers contributed to defective energy distribution to other tissues. Importantly, metabolic stress further aggravated NASH progression in the absence of Fbxw7, resulting from deregulated hepatic and peripheral stress responses mediated by drastic alteration of liver-derived secreted factors known to participate in peripheral glucose and lipid metabolism as well as liver fibrogenesis, including Apoa4, Apcs and Igfbp1. Our findings thus uncover Fbxw7 as a key factor integrating hepatic nutrient and energy-sensing signals to regulate peripheral stress response and NASH progression via the endocrine system, suggesting modulating Fbxw7 function as a strategy to improve systemic metabolism.

The treatment for NAFLD is usually via dietary intervention to achieve weight loss, however lifestyle changes alone are not enough to cease NASH progression, especially for patients undergoing liver inflammation and fibrosis[58], which strongly correlates with transcriptional dysregulation. Nuclear receptors are appealing targets for the development of anti-NASH drugs. Especially, PPARα agonist has previously been reported to alleviate NASH in many aspects, including preventing lipid accumulation, reducing intrahepatic macrophages and hepatic stellate cell to lower liver inflammation and fibrosis, but clinical trials of pure PPARα agonist are disappointing[23]. However, a

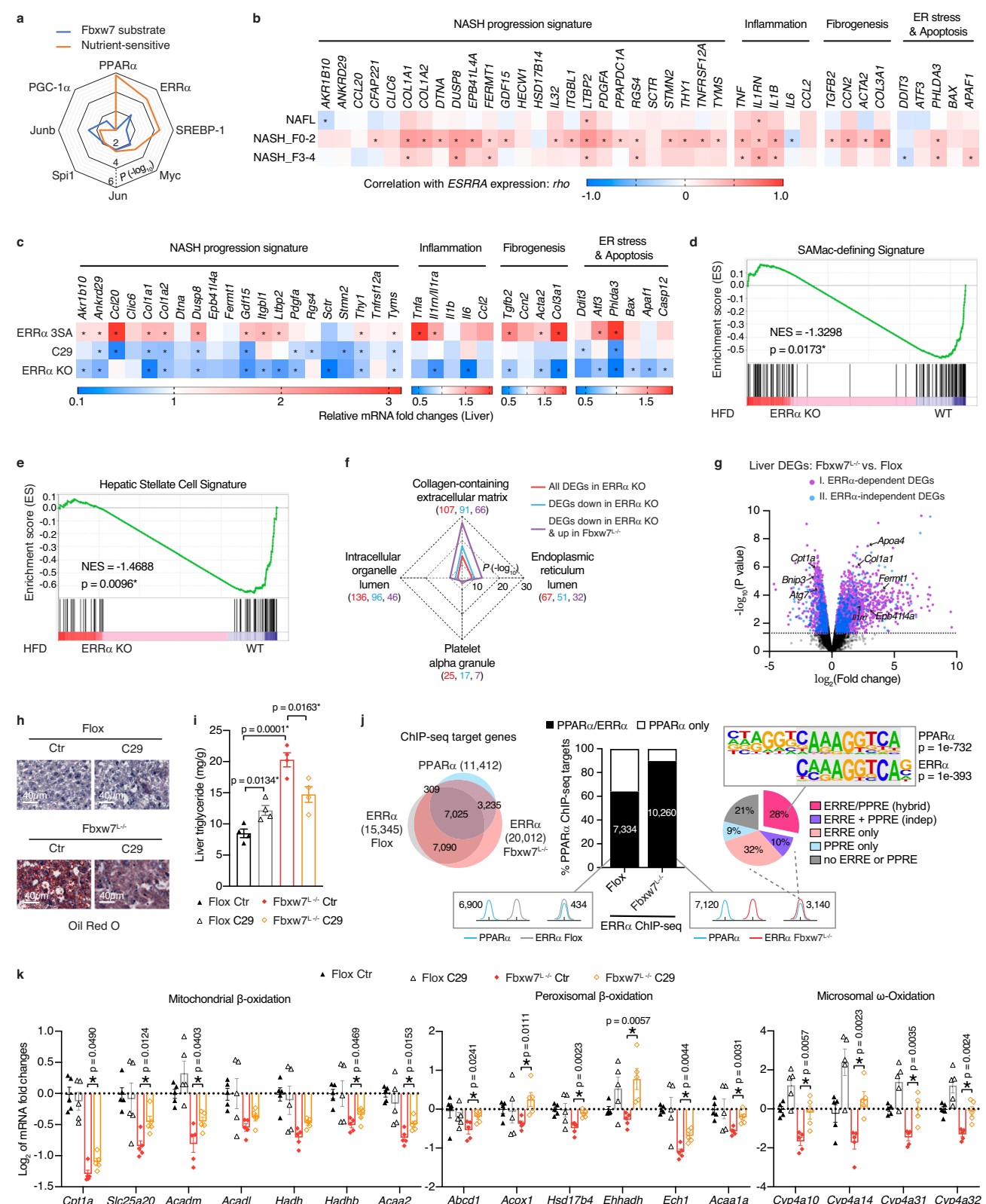

first-in-class pan-PPAR agonist (Lanifibranor) targeting all three PPAR isoforms showed potential as a NASH treatment in a recent clinical trial[39], indicating that the complex etiology of NASH necessitates targeting multiple pathways for successful outcome. Our study reveals positive correlations between *ESRRA* and NASH-related genes in human NAFL/NASH patients. We also uncovered ERRα as a novel regulator of Fbxw7 and demonstrated the potency of inhibiting ERRα

to treat NASH. Although several synthetic ERRα inverse agonists have been developed and have showed benefits in improving whole-body metabolism in obese and diabetic animal models[49,59], there are no current ongoing NAFL/NASH clinical trials targeting ERRα. Our finding that the ERRα cistrome in *Fbxw7*[L–/–] versus *Flox* control mice displays increased overlap with the PPARα cistrome, greatly implies that excess ERRα compete with attenuated PPARα for DNA binding in NASH. Our

**Fig. 8 | Inhibition of Fbxw7 substrate ERRα alleviates NASH. a** Enriched transcription factors downstream of Fbxw7 signaling visualized using -log$_{10}$ (*p*-value). **b** Heatmap representing the *rho* correlation coefficient of *ESRRA* with NASH-related genes in human NAFLD liver biopsies (see Fig. 3g for patient numbers). **c** Heatmap showing RT-qPCR mRNA fold changes of NASH-related genes under a HFD. *n* = 8 for ERRα$^{3SA}$ mice and their littermates; *n* = 5 for ERRα$^{3SA}$ C29 and *n* = 4 for control; *n* = 4 for ERRα KO and WT mice. Upregulation of GSEA SAMac-defining (**d**) and hepatic stellate cell (**e**) signatures in HFD-fed WT mice compared to ERRα KO mice. **f** GO cellular component analysis of DEGs identified in livers of HFD-fed ERRα-null and *Fbxw7$^{L-/-}$* mice (*p* < 0.05, |FC|>1.30). Top enriched terms are presented using -log$_{10}$ (*p*-value). The number of genes per category are indicated in parentheses. **g** Volcano plot of DEGs identified in Fbxw7-null livers (*p* < 0.05, |FC|>1.30). Cluster I is ERRα-dependent DEGs, found differentially bound by ERRα in Fbxw7-null versus *Flox* liver ChIP-seq (±20 kb, |FC|≥1.5); Cluster II is ERRα-independent DEGs that show no differences in ERRα-binding between genotypes (±20 kb, |FC|<1.5). Representative liver oil red O staining (**h**) and hepatic triglyceride levels (**i**, *n* = 4) of *Flox* and *Fbxw7$^{L-/-}$* mice post C29 administration. Scale bars, 40 μm. **j** Overlap of PPARα ChIP-seq targets (±20 kb) with ERRα ChIP-seq targets (±20 kb) identified in control and Fbxw7-null livers (left) and whether the shared targets harbor distinct or overlapping peaks (middle). ERRE and PPRE motif occurrences in 4056 overlapping PPARα and ERRα (*Fbxw7$^{L-/-}$*) peaks at 3140 co-targeted genes whereby 28% harbor an ERRα motif embedded within the PPARα motif (right). **k** RT-qPCR analysis of hepatic genes regulating fat degradation from *Flox* and *Fbxw7$^{L-/-}$* littermates post control or C29 injection, *n* = 5. Data are represented as means ± SEM (**i**, **k**). *\*p* < 0.05, non-parametric Spearman's test (two-tailed) (**b**), unpaired two-tailed Student's *t* test (**c**, **i**, **k**). Source data are provided as a Source Data file including Fig. 8b, c *p*-values.

study thus fosters simultaneous PPARα activation and ERRα inhibition as a potential improved therapeutic option for the treatment of NASH.

## Methods

### Mice

All experiments performed with mice were conducted in accord with accepted procedures by the McGill Facility Animal Care Committee (animal protocol 3173) and complied with the Canadian Council on Animal Care. Unless otherwise noted, mouse experiments involved age-matched male littermates (aged 2–3 months) housed 2–5 per cage. Mice were maintained in a constant environment (humidity: 30–70%; temperature: 18–24 °C; 12 h light/dark cycle: 7AM-7PM light/7PM-7AM dark) with ad libitum access to standard normal diet (ND, cat. no. 2920x; Envigo; Teklad Rodent diet; 3.1 kal/g, 24 kcal% protein, 16 kcal% fat, 60 kcal% carbohydrate) and water in a McGill University animal facility. Mice were sacrificed by cervical dislocation at Zeitgeber time (ZT) 2–4 for serum and tissue collection. Harvested liver, white adipose, and skeletal muscle (gastrocnemius and soleus) were immediately snap-frozen in liquid nitrogen and stored at −80 °C until further processing.

*Fbxw7* floxed mice (cat. no. 017563; The Jackson Laboratory)[60] were obtained on a C57BL/6J genetic background and bred with Alb-Cre mice. Homozygous floxed females were bred with homozygous floxed and hemizygous Cre transgenic males to generate Cre-positive and Cre-negative mice. Primers used for mouse genotyping are listed in Supplementary Data 6. Both ERRα$^{3SA}$ phospho-mutant mice[15] and *Esrra*-null mice[61] on a C57BL/6N genetic background were previously generated.

For mouse fasting studies, mice were transferred into new cages to switch from corn chip to wood chip bedding and fasted for 24 h. For experiments involving refeeding, mice were refed with ND for 2 h post a 22 h fast.

For studies with a ketogenic diet, mice were fed a keto diet paste (cat. no. S3666; Bio-Serv; 7.24 kal/g, 4.7 kcal% protein, 93.5 kcal% fat, 1.8 kcal% carbohydrate) that was replaced every three days for the indicated duration. For HFD studies, WT or ERRα KO mice were fed a HFD (cat. no. TD.06414; Envigo; Teklad Custom diet; 5.1 kal/g, 18.3 kcal% protein, 60.3 kcal% fat, 21.4 kcal% carbohydrate) or a control chow diet (cat. no. TD.08806; Envigo; Teklad Custom diet; 3.6 kal/g, 20.5 kcal% protein, 10.5 kcal% fat, 69.1 kcal% carbohydrate) for a duration of 15 weeks initiated at 6 weeks of age. For HFD studies with WT and ERRα$^{3SA}$ mice, diets were given for 17 weeks.

For administration of the ERRα inhibitor C29[49] to mice, C29 was first dissolved to 125 mg/mL in DMSO (Thermo Fisher Scientific) then diluted (1:25) to 5 mg/mL using a dilution buffer (30% PEG-300 (Sigma), 70% saline). The vehicle control solution was prepared by diluting an equal amount of DMSO with the dilution buffer. C29 was delivered daily by intraperitoneal injection at 15 mg/kg into *Flox* and *Fbxw7$^{L-/-}$* mice for 14 days. For ERRα$^{3SA}$ mice, C29 injections were administered at 20 mg/kg for 21 days after feeding an 18-week HFD.

### RNA isolation, RT-qPCR, and RNA-seq

For total RNA extraction from mouse tissues, a RNeasy Mini Kit (cat. no. 74106; QIAGEN), RNeasy Lipid Tissue Mini Kit (cat. no. 74804; QIAGEN), or a RNeasy Fibrous Tissue Mini Kit (cat. no. 74704; QIAGEN) were used. cDNA was generated from RNA by reverse transcription using ProtoScript II Reverse Transcriptase (cat. no. M0368X; NEB) according to the manufacturer's instructions. cDNA was amplified by real-time quantitative PCR (RT-qPCR) on a LightCycler 480 instrument (Roche) using SYBR Green Master Mix (cat. no. 4887352001; Roche). Relative mouse gene expression levels were normalized to the expression of housekeeping gene *B-actin*, *Hprt*, or *Arbp*. Relative human gene expression levels were normalized to the housekeeping gene *TBP*, *B-ACTIN*, *HPRT1* or *RPLPO*. Sequences of primers used for RT-qPCR are listed in Supplementary Data 6.

For RNA-seq, RNA was prepared as described above and mRNA sequencing and analysis were performed by Novogene Bioinformatics Technology Co. Ltd. Briefly, sequencing libraries were prepared using a NEBNext Ultra RNA Library Prep Kit for Illumina (NEB) following the manufacturer's recommendations. The library preparations were sequenced as 150 bp paired-end reads on an Illumina platform (NovaSeq 6000). STAR software (version 2.5)[62] was employed to align paired-end clean reads to the mouse reference genome mm10 and to count the number of reads per gene. For each gene, FPKM (Fragments per kilobase of transcript sequence per millions base pairs sequenced) was calculated based on the gene length and read counts mapped to the gene. DESeq2 R package (version 2_1.6.3) was used to test for differential expression between two groups whereby genes with *p*-value < 0.05 were considered differentially expressed.

### Cell or tissue lysate preparation and immunoblot analysis

For tissue lysates, frozen mouse tissue powder obtained by pulverization in liquid nitrogen was homogenized and sonicated in buffer K (20 mM Phosphate Buffer pH 7.0, 15 mM NaCl, 1% NP40, 5 mM EDTA) containing protease and phosphatase inhibitors (Thermo Fisher Scientific). Samples were rotated at 4 °C for 1 h followed by centrifugation at 10,000 × *g* for 10 min at 4 °C and the resulting supernatants were transferred into new tubes. For liver nuclear extracts, livers were first homogenized then rotated at 4 °C for 1 h followed by cold centrifugation at 94 × *g*. The isolated nuclear pellets were washed twice with buffer K with subsequent sonication in buffer K and centrifugation. For whole cell extracts of cultured cells, the cells were first washed in ice-cold PBS and lysed using buffer K.

Protein lysates were quantified using the Bradford method with Bio-Rad protein assay dye reagent (cat. no. 5000006). Samples were boiled with SDS sample buffer and denatured proteins were separated by SDS-PAGE and transferred onto PVDF membranes (Bio-Rad). Proteins were blotted according to primary antibody manufacturers' recommendations and detected using ECL Western blotting detection reagents (Amersham Biosciences). Immunoblot images were

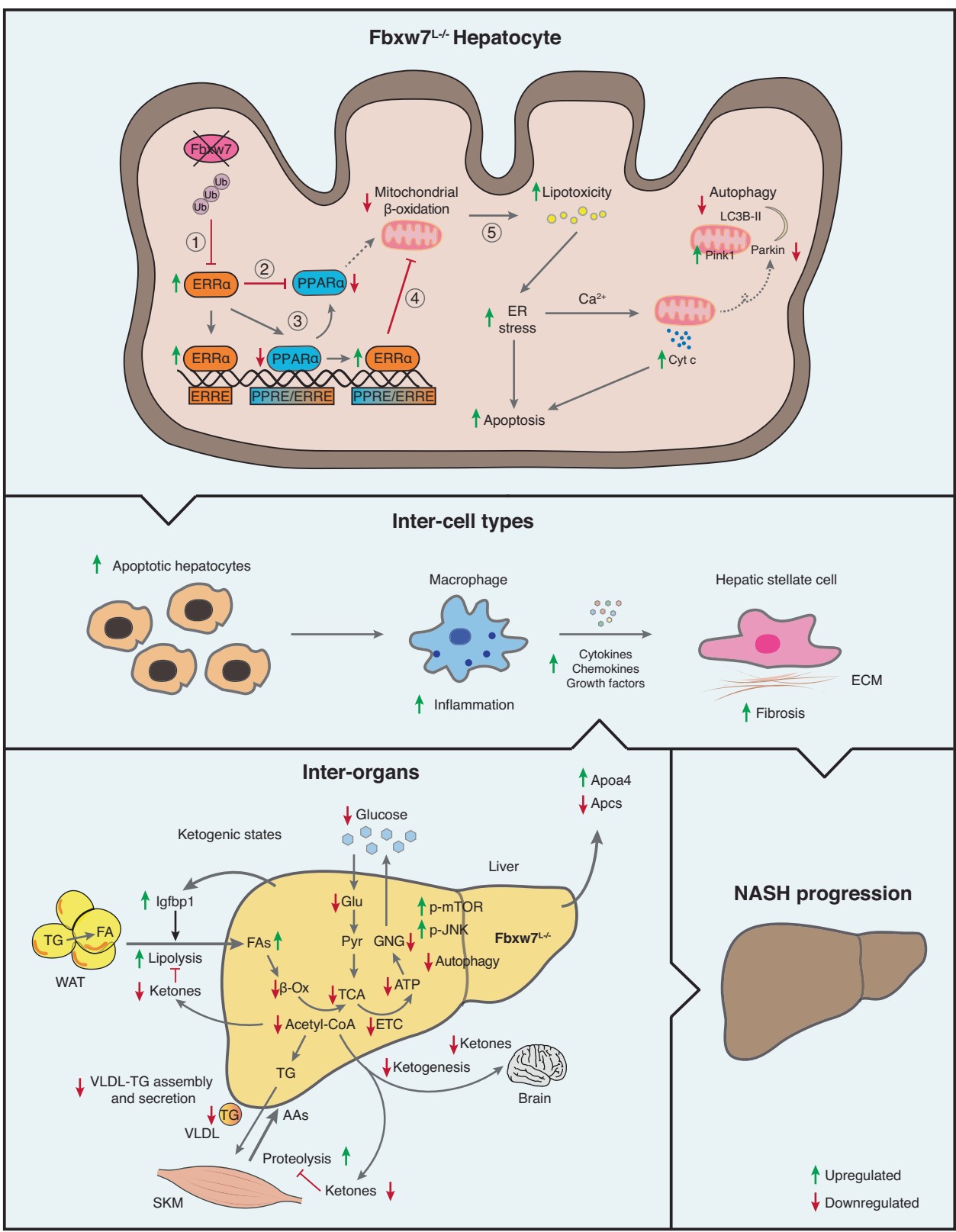

collected using a ChemiDoc MP imaging System (Bio-Rad). Protein bands were quantified using open-source ImageJ software (with Java 1.8.0_172, National Institutes of Health) with levels of phosphorylated proteins normalized to their respective total protein levels and levels of non-phosphorylated proteins normalized to a loading control. Antibodies used in this study are listed in Supplementary Data 7.

## Cell culture and DNA constructs transduction

HEK293T (CRL-3216), HepG2 (HB-8065) and Hepa 1–6 (CRL-1830) cell lines were originally obtained from the American Type Culture Collection (ATCC) and cultured in DMEM (cat. no. 11965092; Thermo Fisher Scientific) supplemented with 10% (v/v) fetal bovine serum (FBS; cat. no. 12483020; Thermo Fisher Scientific), 100 units/mL penicillin-streptomycin (Wisent), and 1X sodium pyruvate (Wisent) at 37 °C in a

**Fig. 9 | Hepatocyte FBXW7 protects against NASH and regulates systemic energy homeostasis by coordinating nutrient-sensing nuclear receptors.** Schematic summary of the mechanisms of NASH progression in the absence of Fbxw7. Hepatocyte Fbxw7 deficiency relieves ERRα degradation (red line) thus elevating ERRα levels (green arrow) ultimately leading to reduced PPARα activity (red arrow). Mechanistically, 1) Loss of Fbxw7 leads to overexpression of ERRα; 2) ERRα transcriptionally represses the expression of PPARα; 3) Elevated levels of ERRα displaces PPARα on shared sites; 4) Combined increased ERRα and decreased PPARα occupancy on shared sites results in inhibition of FAO; 5) Decreased FAO leads to a cascade of cellular stresses including lipotoxicity, increased ER stress,

mitochondrial dysfunction, and decreased autophagy leading to apoptosis, immune infiltration, fibrogenesis and ultimately NASH progression. Investigation of physiological conditions during which Fbxw7-dependent signaling became challenged, such as enhanced hepatic FA influx induced by fasting and a keto diet, further revealed hepatocyte Fbxw7 as a crucial regulator of whole-body stress response and susceptibility to NASH. Cyt c Cytochrome c, ECM extracellular matrix, Glu glucose, Pyr pyruvate, GNG gluconeogenesis. The Figure was partly generated using Servier Medical Art, provided by Servier, licensed under a Creative Commons Attribution 3.0 unported license.

humidified incubator containing 5% $CO_2$. Authentication of HepG2 and Hepa 1–6 cells was based on the expression of insulin-like growth factor II. HEK293T cells were not specifically authenticated beyond being obtained from the ATCC. Cells were routinely screened for possible mycoplasma contamination using a mycoplasma PCR detection kit (cat. no. G238; Applied Biological Materials) and showed no signs of infection.

shFBXW7_A vector (TRCN0000006557), shFBXW7_B vector (TRCN0000006558), and pLX317 v5 tagged FBXW7 vector (TRCN0000477223) were obtained from the TRC3 ORF collections developed by members of the ORFeome Collaboration (Sigma/TransOMIC), provided by the Genetic Perturbation Service of the Rosalind and Morris Goodman Cancer Institute (GCI) at McGill University. pCDNA3.1-PPARA vector (plasmid no. 169019) was purchased from Addgene. Calcium phosphate precipitation was used to transiently transfect HEK293T cells. To obtain stable HepG2 cells overexpressing or silencing FBXW7, DNA vectors were first transfected into HEK 293 cells together with psPAX2 and pMD2.G vectors for virus production. HepG2 cells were infected by incubating in lentivirus-containing media supplemented with 8 μg/mL polybrene and selected with 1 μg/mL puromycin 30 h post infection. siCtr (cat. no. D-001810-10-50; Dharmacon), siFBXW7 (cat. no. L-004264-00-0005; Dharmacon) and siESRRA (cat. no. L-003403-00-0005; Dharmacon) were transfected using the HiPerFect Transfection Reagent (cat. no. 301707; QIAGEN) diluted in Opti-MEM (cat. no. 31985062; Thermo Fisher Scientific).

For drug treatment, Thapsigargin (cat. no. HY-13433; MedChemExpress) and ERRα inhibitor C29[49] were first dissolved in DMSO (Thermo Fisher Scientific) followed by dilution in cell culture media at the indicated concentration to culture cells for the indicated time period. C29 was refreshed every 12 h.

### Immunoprecipitation (IP)

For immunoprecipitation experiments, tissues and cell lysates were prepared using buffer K that was supplemented with 0.3% CHAPS. Equal amounts of protein were incubated with protein A Dynabeads (cat. no. 10008D; Thermo Fisher Scientific) pre-bound with the indicated antibody (0.5 μg/IP). Normal rabbit IgG (cat. no. 10500C; Invitrogen; 0.5 μg/IP) or normal mouse IgG (cat. no. 10400C; Invitrogen; 0.5 μg/IP) were used as a control. After washing with PBS-T (PBS containing 0.1% Tween 20), protein complexes were eluted and denatured in SDS sample buffer in parallel with 2% input samples. Rather than using the conventional HRP-linked rabbit secondary antibody, a special rabbit TrueBlot® IgG (cat. no. 18-8816-33; Rockland; 1:4000 dilution) was used to help eliminate detection of heavy and light chains. Antibodies used for IP are listed in Supplementary Data 7.

### Metabolomics

Metabolomic profiling of hepatic fatty acids, CoAs, amino acids, nucleic acids, glycolysis and TCA cycle intermediates, and other metabolites were carried out by GCI's Metabolomics Innovation Resource (MIR) at McGill University (https://www.mcgill.ca/gci/facilities/metabolomics-innovation-resource-mir). Metabolomics data were normalized to total protein abundance, which was achieved from

a parallel set of replicate samples and quantified via the Bradford protein assay. Missing values were not imputed, and data were rescaled to set the average control group comparison value to 1. See Supplementary Data 1 and Supplementary Data 5 for the metabolomics data summarized in distinct categories with their retention times.

**Reagents.** All LC-MS grade solvents and salts were purchased from Fisher (Ottawa, Ontario, Canada), including water, acetonitrile (ACN), methanol (MeOH), formic acid, ammonium acetate and ammonium formate. The authentic metabolite standards were purchased from Sigma-Aldrich Co. (Oakville, Ontario, Canada).

**Sample extraction.** Snap frozen livers were pulverized on liquid nitrogen and weighed twice ($10 \pm 1$ mg per sample). A volume of 1140 μL of 50% MeOH and 660 μL of ACN was added to each sample prior to bead beating for 2 min at 30 Hz (Eppendorf Tissue-lyser). Lipids were partitioned through the addition of 1800 μL of cold dichloromethane and 900 μL of cold $H_2O$. Samples were kept as cold as possible throughout extraction. The upper aqueous layer was removed and dried by vacuum centrifugation (LabConco) with sample temperature maintained at $-4\,°C$. The lower dichloromethane layer was transferred to glass tubes and dried under nitrogen for GC-MS analysis of free and esterified fatty acids. The dried polar, aqueous layer of each sample was resuspended in 30 μL of cold water and immediately subjected to LC-MS analysis of CoAs followed by GC-MS analysis of polar metabolites (amino acids, nucleic acids, glycolysis and TCA cycle intermediates, other metabolites).

**Free and esterified fatty acids profiling by GC-MS.** To extract FFAs, the dried dichloromethane layer of each sample was dissolved in 1 mL of 80% methanol containing 0.4 M KOH and 1 mL of r $D_{27}$-Myristic acid (internal standard 800 ng/mL). Samples were vortexed and sonicated to aid in resuspension and rest at room temperature for 10 min. After, 1 mL of dichloromethane was added to each sample followed by vortexing and resting for 10 min to allow for phase separation. Samples were then centrifuged for 10 min at $3500 \times g$. The upper basic aqueous layer was removed to a fresh glass tube and acidified by adding 1 mL of 1 M HCl and saturating with NaCl. FFAs were then extracted by the addition of 2 mL hexane and vortexing. Samples were again allowed to rest for 10 min for phase separation followed by centrifugation at $3500 \times g$ for 10 min. The hexane layer was then dried under a stream of nitrogen. FFAs were derivatized by adding 30 μL pyridine and transferred to a glass GC-MS vial insert containing 70 μl N-Methyl-N-tert-butyldimethylsilyltrifluoroacetamide (MTBSTFA). Samples were sealed and incubated at $70\,°C$ for 60 min.

The dichloromethane layer obtained after FFA extraction was dried under a stream of nitrogen. Samples were then resuspended in 1 mL methyl acetate. To generate fatty acid methyl esters, 20 μL of tetramethylammonium hydroxide (25%) were added and allowed to incubate at room temperature with occasional mixing for 4 h. Samples were then dried under a stream of nitrogen. Samples were redissolved in 200 μL hexane and transferred to glass inserts placed in GC-MS vials and sealed. An 80:1 dilution of each sample was also prepared and sealed.

Both tert-butyldimethylsylil-fatty acid (TBDMS-FA) samples and FA methyl esters (FAMES) were analyzed using an Agilent 5975C GC-MS. A sample volume of 1 μL was injected into a Select FAMES (100 m × 250 μm × 0.25 μm) capillary column (Agilent J&W, Santa Clara, CA, USA). The inlet temperature was set to 220 °C, electron impact set at 70 eV and a helium flow rate of 1.3 mL/min (constant flow). Samples were injected in splitless mode and subjected to a temperature gradient starting with 1 min hold at 40 °C followed by a temperature ramp of 20 °C/min to 160 °C, then the ramp was then changed to 1 °C/min to 198 °C and then 5 °C/min to 25 °C and held at 250 °C for 15 min (total run time 70.4 min). Data were collected in scan (50–1000 m/z) mode. After sample data collection the column was baked out of 8 min at 275 °C. Integrated ion intensities were achieved using Mass Hunter Quant software (Agilent technologies). Both TBDMS-FA and FAMES authentic standards were run to confirm mass spectra and retention times.

**Acyl-CoA analysis by LC-MS/MS.** The relative concentrations of acyl-CoAs were measured using a triple quadrupole mass spectrometer (QQQ 6470) equipped with a 1290 ultra high-pressure liquid chromatography system (Agilent Technologies, Santa Clara, California, USA). Chromatographic separation was achieved using a XBridge C18 column 3.5 μm, 2.1 × 150 mm (Waters, USA). The chromatographic gradient started at 100% mobile phase A (10 mM ammonium acetate in water) with a 15 min gradient to 35% B (ACN) at a flow rate of 0.2 mL/min. This was followed by an increase to 100% B in 0.1 min, a 4.9 min hold time at 100% mobile phase B and a subsequent re-equilibration time (6 min) before next injection. A sample volume of 5 μL was injected for each run. Samples were resuspended individually and ran immediately. We ran samples one at a time and rotated between groups to avoid batch effects and to ensure data quality. Authentic standards for CoAs were run after samples, considering CoA instability. Multiple reaction monitoring (MRM) transitions were optimized on standards for each metabolite measured. MRM transitions and retention time windows are summarized in Supplementary Data 8. An Agilent JetStreamTM electro-spray ionization source was used in positive ionization mode with a gas temperature and flow were set at 300 °C and 5 L/min respectively, nebulizer pressure was set at 45 psi and capillary voltage was set at 3500 V. Cell accelerator voltage was set to 4 V for all compounds. Relative concentrations were determined from external calibration curves prepared in water. Ion suppression artifacts were not corrected; thus, the presented metabolite levels are relative to the external calibration curves and should not be considered as absolute concentrations. Data were analyzed using MassHunter Quant (Agilent Technologies).

**Polar metabolites profiling by GC-MS.** The samples were dried by vacuum centrifugation at −4 °C directly after CoA analysis. Samples were resuspended in 30 μL of 10 mg/mL methoxylamine HCl and incubated at room temperature for 30 min. A volume of 70 uL MTBSTFA was added to each sample. Samples were sealed and incubated for 60 min at 70 °C. Immediately after derivatization, samples were subjected to GC-MS analysis. A sample volume of 1 μL was injected into an Agilent 5975 C GC-MS equipped with a DB-5MS + DG (30 m × 250 μm × 0.25 μm) capillary column (Agilent J&W, Santa Clara, CA, USA). All samples were injected three times: twice using scan (50–700 m/z) mode (1× and 15× dilution) and once using selected ion monitoring (SIM) mode. 1 mL of the derivatized sample was injected in the GC in splitless mode with inlet temperature set to 280 °C and electron impact set at 70 eV. Helium, the carrier gas, flow rate was set such that myristic acid eluted at approximately 18 min. The quadrupole was set at 150 °C and the GC-MS interface at 285 °C. The oven program started at 60 °C held for 1 min, then increasing at a rate of 10 °C/min until 320 °C. Bake-out was at 320 °C for 10 min. Integration of ion intensities (generally M-57 ion) was done using the Agilent Mass

Hunter Quant software (Agilent Technologies). Metabolites were identified by comparing mass spectra and retention time to authentic standards run previously at the Goodman Cancer Institute's metabolomics facility.

**ACC activity assay.** Measurement of ACC enzyme activity was conducted using an acetyl-CoA carboxylase assay kit (cat. no. MBS8303295; MyBioSource) that is based on the release of phosphate upon acetyl-CoA carboxylase reacting with acetyl-CoA and ATP. 100 ± 2 mg frozen liver powder per mouse were homogenized with ice-cold assay buffer (1 mL per mg liver) and centrifuged at 8000 × g at 4 °C for 10 min. Supernatants were transferred into new centrifuge tubes and keep on ice for detection. 60 μL supernatant of each sample (or assay buffer for control reactions) were mixed with 40 μL substrate and incubated at 37 °C for 1 h, followed by 5-min boiling. When cold, reactions were centrifuged at 10,000 × g at RT for 5 min. 50 μL of supernatant, water (for blank reaction), or standards (phosphate) were added to 150 μL dye reagent working solution in a 96-well plate, and the absorbance at 635 nm was recorded after 5 min incubation. ACC enzyme activity was normalized to total protein level, with one unit of ACC activity defined as the enzyme generating 1 nmol of $PO_4^{3-}$ per hour.

**Fasn activity assay**

Enzyme activity of Fasn was measured using a spectrophotometry-based assay that monitors the acetyl-CoA- and malonyl-CoA-dependent consumption of the reactant NADPH[63], which has a characteristic absorption peak at 340 nm. For performing the assay, frozen mouse livers were homogenized and sonicated in buffer K, followed by rotation at 4 °C for 1 h and centrifugation at 10,000 × g for 10 min at 4 °C. 20 μL supernatants of each sample were added in duplicate to 160 μL of 0.2 mM NADPH (cat. no. 9000743; Cayman Chemical) in 25 mM Tris pH 8.0 in a 96-well plate and incubated at RT for 10 min with gentle agitation. 20 μL of substrate mixture containing 0.33 mM acetyl CoA (cat. no. 16160; Cayman Chemical) and 1 mM malonyl-CoA (cat. no. 16455; Cayman Chemical), or 20 μL of water (for control wells) were added to the reaction. Absorbance at 340 nm was recorded every 5 min for 30 min. Fasn activity was presented as the rate of decreased absorbance at 340 nm and was normalized to total protein level. Background NADPH oxidation rate in the absence of acetyl-CoA and malonyl-CoA was subtracted for each sample.

**Fatty acid oxidation assay**

FAO activity was measured using a Fatty Acid Oxidation Kit (cat. no. E-141; Biomedical Research Service) that is based on oxidation of the substrate octanoyl-CoA coupled to $FADH_2$/NADH-dependent reduction of INT to formazan. 15 ± 1 mg frozen liver powder per mouse were homogenized with ice-cold 1x sample buffer (50 μL per mg liver) and left on ice with agitation for 5 min followed by centrifugation at 15,000 × g for 5 min. 20 μL supernatants of each sample were added in a 96-well plate in duplicate, mixed with 50 μL control solution and 50 μL reaction solution, and incubated in a 37 °C humidified incubator (no $CO_2$) for 60 min. The colorimetric reaction was read at 492 nm. FAO enzyme activity was normalized to total protein level.

**Biochemistry measurements**

Mouse blood samples were collected from the submandibular vein of mice. Blood glucose and lactate levels were determined using the OneTouch Ultra®2 glucose meter (LifeScan) and the Lactate Scout meter (Lactate.com), respectively. Blood concentrations of ketone bodies were measured using the FreeStyle Precision Neo Blood Glucose and Ketone Monitoring System. Blood was then incubated at RT for 30 min before centrifugation at 2348 × g for serum separation. Serum was stored at −80 °C before use. Serum insulin level was measured using an Ultra Sensitive Mouse Insulin ELISA Kit (cat. no. 90080; Crystal Chem). Serum TG and FFA levels were assessed using a

Triglyceride Assay Kit (cat. no. ab65336; Abcam) and a Free Fatty Acid Assay Kit (cat. no. ab65341; Abcam), respectively, as per the manufacturer's recommendations. Serum L-Amino Acid levels were measured using a L-Amino Acid Assay Kit (ab65347). Liver TG and L-Amino Acid levels were determined using a Triglyceride Assay Kit (cat. no. ab65336; Abcam) and a L-Amino Acid Assay Kit (cat. no. ab65347; Abcam), respectively. Hepatic ATP contents were measured using the StayBrite™ Highly Stable ATP Bioluminescence Assay Kit (catalog no. K791-100; BioVision).

### In vivo adenovirus experiments

Adenoviruses expressing GFP (cat. no. 000541A) or Cre recombinase (cat. no. 000023A) were purchased, amplified, and purified by Applied Biological Materials Inc. $2*10^9$ plaque-forming unit (pfu) adenovirus was diluted in DMEM and delivered to Fbxw7 *Flox* mice via tail vein injection. Mice were sacrificed and livers were collected for analysis 6 days after injection.

### mtDNA/nDNA ratio

The relative number of mitochondria in mouse liver was estimated by measuring the ratio of mtDNA/nDNA. Genomic DNA was extracted using the DNeasy blood and tissue kit (cat. no. 69506; QIAGEN) with RNase A being used to degrade cellular RNA. DNA was quantified and used for qPCR amplification. The relative amounts of mtDNA and nuclear DNA were quantified using primers specific for mitochondrial gene NADH dehydrogenase 1 (*ND1*; forward, 5′-CTAGCAGAAACAAACCGGGC-3′; and reverse, 5′-CCGGCTGCGTATTCTACGTT-3′) and the nuclear gene Hexokinase 2 (*HK2*; forward, 5′-GCCAGCCTCTCCTGATTTTAGTGT-3′; and reverse, 5′-GGGAACACAAAAGACCTCTTCTGG-3′).

### Indirect calorimetry

For simultaneous measurements of whole body metabolic rates, mice were housed individually in metabolic cages (Sable Systems International, Promethion high-definition behavioral phenotyping system) maintained on a 12 h:12 h light:dark cycle at 22 °C with free access to water and food. Mice were fed a ND for two days followed by switching to a keto diet for another three days. $VO_2$, $VCO_2$, RER, locomotor and ambulatory activity, as well as food/water intake were measured after 24 h acclimation. FA oxidation (kcal/h) was assessed by the formula "energy expenditure (kcal/h) x (1-RER)/0.3". CalR (version 1.3)[64] was used to generate hourly plots with gray squares denoting night periods.

### Histology

Liver and eWAT were dissected, fixed in 10% formalin (Thermo Fisher Scientific) and then stored in 70% ethanol at 4 °C. Histological procedures were performed at the Histology Core Facility of the GCI. After embedding in paraffin, tissue fragments were sectioned and stained with hematoxylin and eosin (H&E) or Masson's trichrome (MTC). For immunohistochemistry analysis, liver sections were immunostained with macrophage marker Aif1 (cat. no. 019-19741, FUJIFIILM Wako Pure Chemical Corporation), or hepatic stellate cell/fibroblast marker α-SMA (cat. no. 14-9760-82; Thermo Fisher Scientific). Quantification of immunoreactivity was performed using optimized Aperio digital analysis algorithms. For Oil Red O (ORO) staining, liver sections were embedded in Tissue-Tek O.C.T Compound (cat. no. 1437365; Thermo Fisher Scientific) followed by frozen sectioning and staining. Stained slides were scanned using Aperio ScanScope XT and viewed by Aperio ImageScope (version 12.4.3.5008; Leica Biosystems). White adipocyte cell diameters were measured using Aperio ImageScope, four random sections per mouse were quantified.

### ChIP-qPCR and ChIP-seq

For preparation of liver samples for ChIP assays, livers from two *Flox* or *Fbxw7⁻/⁻* mice fasted for 24 h were pooled, homogenized with a polytron in cold PBS containing protease inhibitors and subsequently cold centrifuged at $845 \times g$. Isolated cell pellets were resuspended in PBS containing 1% formaldehyde with protease inhibitors, rotated at RT for 12 min for DNA-protein crosslinking and subsequently cold centrifuged at $845 \times g$. The cell pellets were washed twice with cold PBS then resuspended in cell lysis buffer (5 mM HEPES pH 8, 85 mM KCl, 0.5% NP-40, protease inhibitors) followed by rotation at 4 °C for 30 min. Nuclear pellets were collected after cold centrifugation at $845 \times g$, resuspended in nuclei lysis buffer (50 mM Tris-HCl pH 8.1, 10 mM EDTA, 1% SDS, protease inhibitors) and subsequently sonicated on ice until the DNA fragments were concentrated between 200–500 base pairs.

For ERRα and PPARα standard ChIP assays, 60 µL of Dynabeads protein G (cat. no. 10003D; Thermo Fisher Scientific) diluted in 250 µL of blocking buffer (PBS, 0.5% BSA, protease inhibitors) were pre-incubated with 10 µL of ERRα antibody (cat. no. ab76228; Abcam) or 4 µL of PPARα antibody (cat. no. MAB3890; Millipore) overnight at 4 °C. Antibody-bound beads were washed 3 times with blocking buffer and incubated overnight at 4 °C with chromatin DNA (200 µg) diluted in 2.5X ChIP dilution buffer (2 mM EDTA pH 8.0, 100 mM NaCl, 20 mM Tris-HCl pH 8.0, 0.5% Triton X-100, protease inhibitors) together with 100 µL of blocking buffer. The next day, beads were washed 3 times with cold LiCl buffer (100 mM Tris-HCl pH 7.5, 500 mM LiCl, 1% NP-40, 1% Na-deoxycholate) each with rotation at 4 °C for 3 min. Beads were then transferred to a new tube and washed twice more with LiCl buffer followed by a brief wash with cold TE buffer (10 mM Tris-HCl pH 7.5, 1 mM EDTA pH 8.0). Then, 300 µL of decrosslink buffer (1% SDS, 0.1 M NaHCO₃) was added to the washed beads and left at 65 °C overnight. Samples were incubated with 300 µL TE buffer and 0.2 µg/µL RNase A (QIAGEN) at 37 °C for 1-2 h followed by incubation at 55 °C with 0.2 µg/µL Proteinase K (BioSHOP) for 1-2 h to digest contaminating RNA and protein. ChIP DNA was purified using a QIAquick PCR Purification Kit (cat. no. 28106; QIAGEN) and eluted with 35 µL of elution buffer (10 mM Tris-HCl pH 8.0, 0.1 mM EDTA pH 8.0). ChIP-qPCR was performed using a LightCycler 480 instrument (Roche) with SYBR Green Master Mix (cat. no. 4887352001; Roche). Relative ChIP enrichment of specific DNA fragments was normalized against two amplified control regions located approximately 4 kb and 49 kb upstream of the ERRα and PROX1 transcriptional start sites, respectively. Sequences of primers used for mouse ChIP-qPCR analyses are listed in Supplementary Data 6.

For ERRα liver ChIP-seq in *Flox* and *Fbxw7⁻/⁻* mice, three individual ChIPs prepared as described above were pooled per ChIP-seq and provided to the Génome Québec Innovation Centre for DNA library preparation using the NEB Ultra II DNA library preparation kit according to Illumina recommendations. The ChIP DNA libraries were sequenced using the NovaSeq 6000 platform (Illumina) as 100 bp paired-end reads. ChIP-seq reads were first trimmed for adapter sequences and low-quality score bases using Trimmomatic v0.36[65]. The resulting reads were mapped to the mouse reference genome (mm10) using BWA-MEM v0.7.12 in paired-end mode at default parameters[66]. Only reads that had a unique alignment (mapping quality > 20) were retained and PCR duplicates were removed using Picard tools v2.0.1 (https://broadinstitute.github.io/picard/). Peaks were called using MACS2 software[67] suite v2.1.1.20160309 at an FDR alpha <0.05 followed by subsequent filtering to FDR < 0.001 using respective sequenced libraries of input DNA as control. Peaks in mitochondrial chromosome and scaffold regions were removed. Peak annotation was performed using the annotatePeaks.pl command from HOMER software suite v4.11.1[68]. Separate "reference peak sets" was generated by merging ChIP-seq peaks across samples in the same experiment, using bedtools merge v2.27.0 with parameters: -sorted -d −150 (https://bedtools.readthedocs.io/). Peak signals (peak summit ± 150 bp) were then calculated as Fragments Per Kilobase of transcript per Million mapped reads (FPKM) using HOMER.

Heatmaps were generated using "computeMatrix" followed by "plotHeatmap" from deepTools v3.5.0 to create binding intensity comparisons at peaks based on normalized (RPKM) bigwig track information[69]. ChIP-seq tracks were visualized using IGV (version v2.8.6)[70]. See Supplementary Data 3 and Supplementary Data 4 for liver ERRα ChIP-seq data from fasted *Flox* control and Fbxw7-null mice, respectively.

PPARα ChIP-sequencing performed on liver of WT male mice treated with PPARα agonist GW7647 was obtained from NCBI's Gene Expression Omnibus (GEO) database: GSM1514930 part of GSE61817 series[40]. Raw PPARα ChIP-seq fastq files were processed using the same pipeline described above for ERRα ChIP and peaks were called using sequenced libraries of PPARα KO liver of GW7647-treated mice as control (GSM1514932)[40].

PPARα and ERRα ChIP-seq peak intersections were performed with HOMER using the script mergePeaks (-d300) and overlapping peaks were annotated using annotatePeaks.pl. TF motif enrichment analysis and instances of motifs near peaks were performed by HOMER using the findMotifsGenome.pl (-size 300) and annotatePeaks.pl (-size 300) commands, respectively. For motif instances near peaks, HOMER motif files for ERRα (erra.motif), ERRβ (esrrb.motif), ERRγ (errg.motif) were used to define the presence of an ERRE and motif files for PPARα (ppara.motif) and PPARγ (pparg.motif) were used to define the presence of a PPRE.

For standard ERRα ChIP experiments in Hepa 1–6 cells, cells cultured in 15 cm plates were fixed with formaldehyde (1% final concentration) at RT for 10 min. Cells were then lysed and sonicated, and DNA-protein complexes were enriched by immunoprecipitation using the above protocol.

### Bioinformatic analyses

Raw RNA-seq data of Fbxw7-null and *Flox* livers (ZT1) were obtained from NCBI Sequence Read Archive (SRA ID: SRP059440)[17], using the command fastq-dump in the SRA Toolkit (https://github.com/ncbi/sra-tools). Adapter sequences and low-quality score bases (Phred score <30) were first trimmed using Trimmomatic v0.36[65]. The resulting reads were aligned to the mouse genome reference (mm10) using STAR software (version 2.5)[62]. Read counts are obtained using HTSeq[71] and are represented as a table which reports, for each sample (columns), the number of reads mapped to a given gene (rows). For all downstream analyses, we excluded lowly-expressed genes with an average read count lower than 10 across all samples. The R package limma[72] was used to identify differences in gene expression levels between genotypes. Nominal p-values were corrected for multiple testing using the Benjamini-Hochberg method. Genes with $p < 0.05$, $|FC| > 1.30$ in $Fbxw7^{L-/-}$ vs *Flox* liver ($n = 3$) were considered significantly dysregulated. Enrichr[73] was used for functional enrichment of DEGs. PCA plot was done with Phantasus v1.17.4 (https://artyomovlab.wustl.edu/phantasus/). GSEA (version 4.2.3) was performed as previously published[74,75]. Hallmark signatures of inflammatory response, hepatic fibrosis, and apoptosis were derived from the MSigDB (version 7.5.1). CIBERSORT[76] algorithm was used to produce the absolute immune fraction score that quantitatively measures the overall immune content, estimated by the median expression level of all genes in the LM22 signature matrix divided by the median expression level of all genes in *Flox* or Fbxw7-null livers. EPIC[77] algorithm was employed for estimating the mRNA fractions of specific cell types from bulk liver gene expression data of *Flox* and $Fbxw7^{L-/-}$ mice. Analyses of *FBXW7*, *PPARA* gene expression and correlation in normal and HCC liver tissues were performed using TNMplot[78] based on RNA-seq from The Cancer Genome Atlas (TCGA), Therapeutically Applicable Research to Generate Effective Treatments (TARGET), and The Genotype-Tissue Expression (GTEx)

repositories. Survival analysis of 364 HCC patients was conducted using Kaplan–Meier Plotter[79], with the cutoff value determining "High" and "Low" groups selected as the best performing threshold among all possible cutoff values computed. For the identification of liver enriched secreted factors, we examined the mRNA expression profile of genes annotated to encode secreted proteins in a panel of 12 mouse tissues (GSE54650)[80], and a cluster of 289 genes exhibiting over 1.5 fold enriched expression in liver compared with the mean expression of the 12 examined tissues was considered as liver-enriched.

### Statistics and reproducibility

GraphPad Prism 9 software and Microsoft Excel (version 16.16.27) were used to generate graphs and for statistical analyses. For indirect calorimetry studies, F-test for analysis of covariance (ANCOVA) using body weight as covariate and one-way analysis of variance (ANOVA) were performed using CalR (version 1.3). The D'Agostino-Pearson omnibus normality test was used to determine the normal distribution. Significant differences were defined as *$p < 0.05$ with the specific statistical tests used and definition of error bars denoted in the figure legends. The value of "n" for biological replicates represents the number of individual mice for in vivo experiments or independent biological replicates for cultured cell experiments. Sample sizes were not pre-specified statistically. For all in vivo experiments, at least three biological replicates were employed unless otherwise indicated. Panels shown without biological replicates are representative of independent experiments.

### Reporting summary

Further information on research design is available in the Nature Portfolio Reporting Summary linked to this article.

## Data availability

ERRα ChIP-sequencing data performed on livers from *Flox* and $Fbxw7^{L-/-}$ mice as well as liver RNA-sequencing data of WT and ERRα KO mice fed a chow or a HFD for 15 weeks have been deposited in NCBI's Gene Expression Omnibus (GEO) and are accessible through GEO SuperSeries accession number GSE205847 encompassing SubSeries GSE205845 (ChIP-seq) and GSE205846 (RNA-seq). Raw RNA-seq data of Fbxw7-null and *Flox* livers (ZT1) were obtained from NCBI Sequence Read Archive (SRA ID: SRP059440). Raw PPARα liver ChIP-seq data of GW7647-treated WT or PPARα KO mice as control were obtained from NCBI Sequence Read Archive (SRA ID: SRP047534). Reference mouse genome mm10 downloaded from the genome website browser (http://hgdownload.cse.ucsc.edu/goldenpath/mm10/bigZips/) was used for data analysis. Human NAFL/NASH and fibrotic liver biopsies RNA-seq GEO: GSE135251; cirrhotic human liver microarray GEO: GSE6764; gene expression in a panel of 12 mouse tissues GEO: GSE54650; fasted mouse liver RNA-seq GEO: GSE46495; hepatic profile post a keto diet GEO: GSE7699; and ERRα mouse liver ChIP-seq data GEO: GSE43638 used in this study were downloaded from GEO. This paper does not report original code. Source data values as well as uncropped immunoblots and gels are provided with this paper as Source Data files. Any additional information required to reanalyze the data reported in this paper is available from the lead contact. Source data are provided with this paper.

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

## Acknowledgements

We thank members of the Giguère laboratory for their support and helpful discussions; Dr. Daina Avizonis and Mrs. Mariana Russo from the MIR at the GCI for their assistance with GC-MS and LC-MS studies; the Histology Core Facility at the McGill GCI for assistance with processing tissue samples; Mr. Carlo Ouellet for help with mouse husbandry, in vivo adenovirus injection and mouse dissections; Ms. Bozena Samborska in Dr. Lawrence Kazak laboratory for assistance with studies involving indirect calorimetry; Dr. Arnim Pause for providing the Alb-Cre mice; Mr. Gerardo Zapata and Dr. Alain Pacis from the Canadian Centre for Computational Genomics (C3G) for help with ChIP-seq analysis and re-analysis of the published raw RNA-seq data of Fbxw7-null livers. This work was supported by a Foundation Grant from the Canadian Institutes of Health Research (FDT-156254) and a Program Project Grant from the Terry Fox Research Institute to V.G. H.X. received a Charlotte and Leo Karassik Foundation Oncology PhD Fellowship, a Fonds de Recherche du Québec-Santé (FRQS) Doctoral award, and a Victor K S Lui Fellowship; Y.M. is supported by a FRQS post-doctoral fellowship; C.S. is a recipient of a Canderel Fellowship; W.B. was supported in part by the McGill Integrated Cancer Research Training Program.

## Author contributions

H.X. and V.G. conceived the study and developed the methodology; H.X., C.R.D., Y.M., C.S., Y.C., C.G., and W.B. performed the experimental work; H.X. and C.R.D. analyzed the data; H.X. wrote the original draft of the manuscript with editing from C.R.D. and V.G.; V.G. supervised the project and acquired funding. All authors approved the manuscript.

## Competing interests

The authors declare no competing interests.
