## [Peer Review File · Nature Communications]

Hepatocyte FBXW7-dependent activity of nutrient-sensing nuclear receptors controls systemic energy homeostasis and NASH progression in male miceREVIEWER COMMENTS

Reviewer #1 (Remarks to the Author):

Xia et al present a manuscript characterizing liver specific deletion of genes encoding the FBXW7- an ubiquitin proteasome system that controls protein degradation- in the context of non-alcoholic steatohepatitis (NASH). Previous works demonstrated that the deletion of FBXW7 in the liver leads to NASH like liver disease, steatohepatitis and impressive accumulation of TAGs (PMID: 21123947). In this manuscript authors, attempt to dive deeper into the phenotype, integrating various -omics analysis, analysis of multiple organs and indicate a potential function of nutrient sensing nuclear receptors ERRa and PPARa during the progression to NASH. While the manuscript embodies a substantial amount of data, some conclusions are not yet supported by the data:

Major points:

- Authors normalize all relative abundances/fold changes from metabolomics measurements to tissue weight. While for most tissues this is a common practice, for the liver that accumulates a huge amount of TAG, this may not be the ideal. TAG as shown in the manuscript, increase by around 3-fold (Supplementary Fig 2a), thus in fatty livers, TAGs become one of the dominant component of the weight. Normalization of the metabolomics data to total liver weight that is constituted in large portion from fat can give a misleading results and inaccurate conclusions. Indeed, most of the metabolomics data showed in this manuscript present decreasing levels of metabolites in FBXW7LKO livers, with the exception of lipids. In this circumstances, where liver mass increased by huge accumulation of lipids, normalization to total protein level would be more suitable.

- Malonyl-CoA static level in manuscript is shown to decrease. Malonyl-CoA is central for de novo lipogenesis (DNL), but it's also an inhibitor of CPT1 that regulates the entry of fatty acids into the mitochondria for B-oxidation. Thus low Malonyl-CoA leads to low DNL but also increased Ox (and vice versa), while in manuscript author argue that both DNL and B-oxidation are down. To ensure that static metabolomics measurement are actually accurate, functional analysis to measure B-oxidation (e.g. stable isotope metabolomics of ¹³C- Octanoate; ¹⁴C-Palmitate oxidation, PMID: 24862277; or fatty acid oxidation kit) and/or DNL (e.g. labeled acetate incorporation in lipids, either stable or radioactive; administration of heavy water in vivo and measure of deuterated FA (PMID: 25271296); measure activity of FAS) could support these results.

- It is unclear how authors perform part of the LC-MS/MS metabolomics analysis (Fig 2e, Supplementary 6 k), are those putative metabolites? How were they extracted, separated and identified. In methods section, authors described only FA and Acyl-CoA analysis.

- Figure 6 is a little confusing. Are the switches between normal diet to ketogenic diet on mice post 20 days on a ketogenic diet or these are acute interventions? If FBXW7LKO mice on ketogenic diet loss weight, but they had the same energy intake, energy expenditure and physical activity (page 16), where does the energy go? For a purpose of transparency, mice body weight (g) as well as food consumption (kcal/day/mouse) should be presented as a timeline. Especially with ketogenic diet, the body weight and food consumption could vary based on the duration of the diet where there is an adaptation period plus ketogenic diet could decrease appetite over time (PMID: 25698989). One data point (Supplementary 6 i) in this case for energy intake is too simplistic to draw any conclusions.

Minor points:

- Supplementary 6 k and text on page 16/17. Authors state: "A keto diet triggered liver fat accumulation, especially in Fbxw7LKO, accompanied by.... reduction in hepatic ketogenesis intermediates" but the data that authors present does not show it. AcAc-CoA and HMG-CoA are not reduced in Fbxw7LKO based on the presented data. Authors also have to keep in mind that while both of these metabolites are the substrates for ketones in the mitochondria, they are also contributing to mevalonate pathway and cholesterol production in the cytosol.

- Fig 7f and text on page 18. Bdh2 is not robustly involved in mitochondrial ketone body metabolism or ketogenesis. Bdh2 is a cytoplasmic version of Bdh1, with only 20% of sequence identity to BDH1, with a high Km for ketone bodies (~10 mM), and governs iron homeostasis (PMID: 34633859).

Reviewer #2 (Remarks to the Author):

In this manuscript, Xia and the colleagues investigate the function of liver Fbxw7 in regulating systemic fatty acids homeostasis and contributing to NASH development. Overall, the results are convincing, and the manuscript is well written.

Major:

Please add the line number so it makes the reviewers easy to quote the sentences.

1c 1d. The author used the overlapped genes for the analysis, but 1) the NASH F4 is very late and severe stage, and the transcriptome at this time point may be composed of compensatory responses and cannot reflect the causal ones, and 2) the mouse and human RNA-seq data are from independent studies and the author also used different criteria to define the DEGs, thus the number of the overlapped genes is very artificial.

They should be moved to supplementary, and here in the main figure the mouse liver RNA results should be enough and better.

In addition, the $|FC| > 1.30$ with $p < 0.05$ is too mild to define the DEGs. I suggest at least to use $|FC| > 1.5$ and $FDR < 0.05$, and also GO of up- and down-regulated genes should be separated.

1f. Fbxw7 should also be included to the WB to show the KO efficiency.

1g 1h. Staining of marks of intrahepatic macrophage, fibroblast and hepatic stellate cells in Fbxw7 Flox and LKO livers should be included, and this would be most direct and solid than the predicted results 1g-1j.

2. How the liver weight changes after deletion of Fbxw7?

2g,2h. Whether the expression of SREBP1c (mRNA and protein) changes after deletion of Fbxw7?

2l. How is the blood insulin level?

3a. To better show the quality of the RNAseq from Fbxw1 KO and Flox livers, the author should have a PCA plot in supplementary. In addition, it is very misleading if only use $p < 0.05$ to define the significantly changed genes. Please use at least $|FC| > 1.5$ and $FDR < 0.05$ to find DEGs.

3l. Same as 1f, Fbxw7 should also be plotted to show the KO efficiency.

7a. Since the author has the RNAseq data, why still use RT-PCR results for the heatmap, which is less reliable.

8b. It would be better to add the correlation heatmap with FBXW7 and PPARA besides ESRRA.

8k. Remove the motifs figure as well as the related sentences in the main text since they are wrong. These two ERRE are actually both half NR binding site, while the top PPRE seems not any NR motif and the bottom PPRE is actually a full NR (DR1) binding (which can be the motif of any DR1 NR).

Minor:

1 m&n, Usually the over-expressed protein level is much higher than the physiological level, so this result makes no big sense for me. Although it fits your hypothesis, I suggest to move it to supplementary figs.

3a. Please include the number of mice/livers used for RNAseq.

4a. Please also indicate the number of genes identified in each GO CC.

Reviewer #3 (Remarks to the Author):

This is a very elegant and highly comprehensive work on the role of FBXW7 in the development of pathogenesis of NAFLD and NASH using a combination of diverse approaches. They found that reduced levels of FBXW7 and PPARA as a key feature of advanced NASH. Interestingly, they demonstrated that loss of Fbxw7 in hepatocyte downregulates the almost entire fat degradation network without affecting the lipogenic program, which is critical for NAFLD initiation but is unlikely to be the main contributor to NASH progression. Their results convincingly show that the accumulated TG in Fbxw7-null livers is predominantly caused by their massive defects in fat clearance associated with attenuated PPARA expression. NASH. They further show that inhibition of FBXW7 substrate ERRA alleviates NASH via competition for the same genomic binding sites that recruits PPARA. Overall, this study suggest that simultaneous PPARA activation and ERRA inhibition as a potential therapeutic NASH treatment.

While this study is comprehensive, it is overall too descriptive and lacks mechanistic insights. Most importantly, while PPARA is a key player in the action of FBXW7, the molecular details on how FBXW7 controls the activity of PPARA either directly or indirectly are completely missing. Additional experiments showing this missing link will make this work stronger and more insightful and complete.

With adding/addressing the major mechanistic studies as suggested above, this reviewer recommends publication of this beautiful work in Nature Communications.

NCOMMS-22-25875-T: Response to reviewers' comments

Reviewer 1:

Xia et al present a manuscript characterizing liver specific deletion of genes encoding the FBXW7- an ubiquitin proteasome system that controls protein degradation- in the context of non-alcoholic steatohepatitis (NASH). Previous works demonstrated that the deletion of FBXW7 in the liver leads to NASH like liver disease, steatohepatitis and impressive accumulation of TAGs (PMID: 21123947). In this manuscript authors, attempt to dive deeper into the phenotype, integrating various -omics analysis, analysis of multiple organs and indicate a potential function of nutrient sensing nuclear receptors ERRa and PPARa during the progression to NASH. While the manuscript embodies a substantial amount of data, some conclusions are not yet supported by the data:

We thank the reviewer for carefully reviewing our manuscript and the constructive comments. The reviewer felt that some conclusions were not yet supported by the data, and while we initially disagreed with that statement, we have now performed a large number of experiments to further support our conclusions that essentially remain the same. Our study aimed to reveal the significance and mechanisms underlying Fbxw7-dependent regulation of NASH progression, a disease for which the underlying causes and molecular mechanism remained to be elucidated. This work and its conclusions are an important contribution toward this goal and for the future development of improved anti-NASH therapy. We have thus carefully addressed all the points raised by the reviewer which significantly reinforced key findings of the study. Please see our detailed response below.

1. Authors normalize all relative abundances/fold changes from metabolomics measurements to tissue weight. While for most tissues this is a common practice, for the liver that accumulates a huge amount of TAG, this may not be the ideal. TAG as shown in the manuscript, increase by around 3-fold (Supplementary Fig 2a), thus in fatty livers, TAGs become one of the dominant component of the weight. Normalization of the metabolomics data to total liver weight that is constituted in large portion from fat can give a misleading results and inaccurate conclusions. Indeed, most of the metabolomics data showed in this manuscript present decreasing levels of metabolites in FBXW7LKO livers, with the exception of lipids. In this circumstances, where liver mass increased by huge accumulation of lipids, normalization to total protein level would be more suitable.

Response: We thank the reviewer for pointing this out. Normalization challenges result from the lack of a proper standard method for direct measurement of the total metabolites, which depends on several factors such as sample weight and cellular masses. We agree with the reviewer that it is more suitable to normalize metabolite levels to total protein levels. The general advantage of this strategy is the high linear correlation between metabolite abundance and protein amount in the tissue, which permits comparisons between samples with different cellular masses, such as in the case of fatty liver. Considering poor protein recovery in metabolomics-compatible solvents and incomplete protein re-solubilization from the remaining cell pellet, we measured total protein amounts using a parallel set of replicate samples via the Bradford protein assay and re-normalized our metabolomics dataset to protein content. Accordingly, all associated figures (**Fig. 2b, e, Fig. 3c, Supplementary Fig. 2f, Supplementary Fig. 6k, Supplementary Fig. 7l**) and **Supplementary Data 1 and 3** have been updated using this approach. Importantly, this had no

significant impact on our findings. In addition to fatty acids, metabolites like asparagine, GABA, β -alanine, and uracil were also upregulated, thus we think the decrease observed in metabolites in FBXW7 LKO was certainly not caused by the normalization method used in the original manuscript.

2. Malonyl-CoA static level in manuscript is shown to decrease. Malonyl-CoA is central for *de novo* lipogenesis (DNL), but it's also an inhibitor of CPT1 that regulates the entry of fatty acids into the mitochondria for β -oxidation. Thus low Malonyl-CoA leads to low DNL but also increased β -oxidation (and vice versa), while in manuscript author argue that both DNL and β -oxidation are down. To ensure that static metabolomics measurement are actually accurate, functional analysis to measure β -oxidation (e.g. stable isotope metabolomics of ^{13}C - Octanoate; ^{14}C - Palmitate oxidation, PMID: 24862277; or fatty acid oxidation kit) and/or DNL (e.g. labeled acetate incorporation in lipids, either stable or radioactive; administration of heavy water *in vivo* and measure of deuterated FA (PMID: 25271296); measure activity of FAS) could support these results.

Response: In a normal physiological state, increased *de novo* lipogenesis (DNL) is usually accompanied by decreased fatty acid oxidation (FAO). However, it may not be the case in a pathological state. Both DNL and FAO were found upregulated upon HFD in multiple studies (e.g. PMID: 30893613). NASH is a dynamic disease with constant expanding and declining steatohepatic activity, as well as disrupted balance between DNL and FAO.

Hepatic FAO activity is regulated at multiple nodes of regulation including the effects of malonyl-CoA, which plays a central role in lipid handling acting as both an essential substrate in DNL and an inhibitor of FAO via repressing the FAO rate-limiting enzyme CPT1a. Fbxw7 LKO mice exhibit significantly lower malonyl-CoA levels, a strong indicator of suppressed DNL. Although the lower malonyl-CoA levels could simultaneously lead to increased FAO via de-repression of CPT1a, Fbxw7 LKO mice display significantly reduced mRNA and protein levels of Cpt1a (**Fig. 3e, k**), thus lacking the ability to augment FAO via Cpt1a activation. Additionally, malonyl-CoA-independent regulation of FAO may be particularly important in the liver, as the liver CPT1a isoform is less sensitive to malonyl-CoA inhibition than its heart and muscle counterpart CPT1b (PMID: 10651636). As suggested, we further support our metabolic phenotype as evidenced by the measurement of 1) ACC (acetyl-CoA carboxylase) enzyme activity responsible for converting acetyl-CoA to malonyl-CoA; 2) Fasn (fatty acid synthase) that catalyses the conversion of malonyl-CoA with acetyl-CoA to generate palmitate, and 3) FAO activity. We also detailed these measurements in the **Methods** section. As presented in **New Fig. 2i, j, and New Fig. 3d**, Fbxw7-null liver displayed similar ACC and Fasn activities but remarkably reduced hepatic FAO activity as compared to Flox controls. Thus, reduced malonyl-CoA and acetyl-CoA levels in Fbxw7-null liver are mainly contributed by attenuated FAO, supported by their decreased hepatic fatty acyl-CoA content (**Fig. 3c**). Furthermore, loss of Fbxw7 resulted in the drastic reduction of hepatic CoA abundance concomitant with downregulated CoA biosynthesis genes required for fatty acid activation and acetyl-CoA production as shown in **New Supplementary Fig. 2f, g**.

3. It is unclear how authors perform part of the LC-MS/MS metabolomics analysis (Fig 2e, Supplementary 6 k), are those putative metabolites? How were they extracted, separated and identified. In methods section, authors described only FA and Acyl-CoA analysis.

Response: We apologize for the confusion. The metabolites presented in Fig. 2e and Supplementary 6k are not putative and were directly measured from our metabolomics study performed at the Goodman Cancer Institute's metabolomics core facility. All measured metabolites were validated against authentic standards. For GC-MS this means retention time and mass spectrum, for LC-MS it is retention time and multiple reaction monitoring. We have updated our **Methods** section to clarify this as presented below.

Metabolomics

Metabolomic profiling of hepatic fatty acids, CoAs, amino acids, nucleic acids, glycolysis and TCA cycle intermediates, and other metabolites were carried out by GCI's Metabolomics Innovation Resource (MIR) at McGill University (<https://www.mcgill.ca/gci/facilities/metabolomics-innovation-resource-mir>). Metabolomics data were normalized to total protein abundance, which was achieved from a parallel set of replicate samples and quantified via the Bradford protein assay. See Supplementary Data 1 and Supplementary Data 3 for the metabolomics data summarized in distinct categories with their retention times.

Reagents: All LC-MS grade solvents and salts were purchased from Fisher (Ottawa, Ontario, Canada), including water, acetonitrile (ACN), methanol (MeOH), formic acid, ammonium acetate and ammonium formate. The authentic metabolite standards were purchased from Sigma-Aldrich Co. (Oakville, Ontario, Canada).

Sample extraction: Snap frozen livers were pulverized on liquid nitrogen and weighed twice (10 ± 1 mg per sample). A volume of 1140 μ L of 50% MeOH and 660 μ L of ACN was added to each sample, and then subjected to bead beating for 2 min at 30 Hz (Eppendorf Tissue-lyser). Lipids were partitioned through the addition of 1,800 μ L of cold dichloromethane and 900 μ L of cold H₂O. Samples were kept as cold as possible during extraction. The upper aqueous layer was then removed and dried by vacuum centrifugation (LabConco) with sample temperature maintained at -4°C. The lower dichloromethane layer was transferred to glass tubes and dried under nitrogen for GC-MS analysis of free and esterified fatty acids. The dried polar, aqueous layer of each sample was resuspended in 30 μ L of cold water and immediately subjected to LC-MS analysis of CoAs followed by GC-MS analysis of polar metabolites (amino acids, nucleic acids, glycolysis and TCA cycle intermediates, other metabolites).

Free and esterified fatty acids profiling by GC-MS: To extract FFAs, the dried dichloromethane layer of each sample was dissolved in 1 mL of 80% methanol containing 0.4 M KOH and 1 ml of r D₂₇-Myristic acid (internal standard 800 ng/ml). Samples were vortexed and sonicated to aid in resuspension and rest at room temperature for 10 min. After, 1 ml of dichloromethane was added to each sample followed by vortexing and resting for 10 min to allow for phase separation. Samples were then centrifuged for 10 min at 3,500 g. The upper basic aqueous layer was removed to a fresh glass tube and acidified by adding 1 ml of 1 M HCl and saturating with NaCl. FFAs were then extracted by the addition of 2 mL hexane and vortexing. Samples were again allowed to rest for 10 min for phase separation followed by centrifugation at 3,500 g for 10 min. The hexane layer was then dried under a stream of nitrogen. FFAs were derivatized by adding 30 µL pyridine and transferred to a glass GC-MS vial insert containing 70 µl N-Methyl-N-tert-butyltrimethylsilyltrifluoroacetamide (MTBSTFA). Samples were sealed and incubated at 70°C for 60 min.

The dichloromethane layer obtained after FFA extraction was dried under a stream of nitrogen. Samples were then resuspended in 1 mL methyl acetate. To generate fatty acid methyl esters, 20 µL of tetramethylammonium hydroxide (25%) were added and allowed to incubate at room temperature with occasional mixing for 4 h. Samples were then dried under a stream of nitrogen. Samples were redissolved in 200 µL hexane and transferred to glass inserts placed in GC-MS vials and sealed. An 80:1 dilution of each sample was also prepared and sealed.

Both tert-butyltrimethylsilyl-fatty acid (TBDMS-FA) samples and FA methyl esters (FAMES) were analyzed using an Agilent 5975C GC-MS. A sample volume of 1 µL was injected into a Select FAMES (100 m x 250 µm x 0.25 µm) capillary column (Agilent J&W, Santa Clara, CA, USA). The inlet temperature was set to 220°C, electron impact set at 70 eV and a helium flow rate of 1.3 mL/min (constant flow). Samples were injected in splitless mode and subjected to a temperature gradient starting with 1 min hold at 40°C followed by a temperature ramp of 20°C/min to 160°C, then the ramp was then changed to 1°C/min to 198°C and then 5°C/min to 250°C and held at 250°C for 15 min (total run time 70.4 min). Data were collected in scan (50-1000 m/z) mode. After sample data collection the column was baked out of 8 min at 275°C. Integrated ion intensities were achieved using Mass Hunter Quant software (Agilent technologies). Both TBDMS-FA and FAMES authentic standards were run to confirm mass spectra and retention times.

Acyl-CoA analysis by LC-MS/MS: The relative concentrations of acyl-CoAs were measured using a triple quadrupole mass spectrometer (QQQ 6470) equipped with a 1290 ultra high-pressure liquid chromatography system (Agilent Technologies, Santa Clara, California, USA). Chromatographic separation was achieved using a XBridge C18 column 3.5 µm, 2.1×150 mm (Waters, USA). The chromatographic gradient started at 100% mobile phase A (10 mM ammonium acetate in water) with a 15 min gradient to 35% B (ACN) at a flow rate of 0.2 ml/min. This was followed by an increase to 100% B in 0.1 min, a 4.9 min hold time at 100% mobile phase B and a subsequent re-equilibration time (6 min) before next injection. A sample volume of 5 µL was injected for each run. Samples were resuspended individually and ran immediately. We ran samples one at a time and rotated between groups to avoid batch effects and to ensure data quality. Authentic standards for CoAs were run after samples, considering CoA instability. Multiple reaction monitoring (MRM) transitions were optimized on standards for each metabolite measured. MRM transitions and retention time windows are summarized in Supplementary Data 6. An Agilent JetStream™ electro-spray ionization source was used in positive ionization mode with a gas temperature and flow were set at 300°C and 5 L/min respectively, nebulizer pressure

was set at 45 psi and capillary voltage was set at 3500 V. Cell accelerator voltage was set to 4 V for all compounds. Relative concentrations were determined from external calibration curves prepared in water. Ion suppression artifacts were not corrected; thus, the presented metabolite levels are relative to the external calibration curves and should not be considered as absolute concentrations. Data were analyzed using MassHunter Quant (Agilent Technologies).

Polar metabolites profiling by GC-MS: The samples were dried by vacuum centrifugation at -4°C directly after CoA analysis. Samples were resuspended in 30 µL of 10 mg/ml methoxyamine HCl and incubated at room temperature for 30 min. A volume of 70 µL MTBSTFA was added to each sample. Samples were sealed and incubated for 60 min at 70°C. Immediately after derivatization, samples were subjected to GC-MS analysis. A sample volume of 1 µL was injected into an Agilent 5975C GC-MS equipped with a DB-5MS+DG (30 m x 250 µm x 0.25 µm) capillary column (Agilent J&W, Santa Clara, CA, USA). All samples were injected three times: twice using scan (50-700 m/z) mode (1x and 15x dilution) and once using selected ion monitoring (SIM) mode. 1 mL of the derivatized sample was injected in the GC in splitless mode with inlet temperature set to 280°C and electron impact set at 70 eV. Helium, the carrier gas, flow rate was set such that myristic acid eluted at approximately 18 min. The quadrupole was set at 150°C and the GC-MS interface at 285°C. The oven program started at 60°C held for 1 min, then increasing at a rate of 10°C/min until 320°C. Bake-out was at 320°C for 10 min. Integration of ion intensities (generally M-57 ion) was done using the Agilent Mass Hunter Quant software (Agilent Technologies). Metabolites were identified by comparing mass spectra and retention time to authentic standards run previously at the Goodman Cancer Institute's metabolomics facility.

4. Figure 6 is a little confusing. Are the switches between normal diet to ketogenic diet on mice post 20 days on a ketogenic diet or these are acute interventions? If FBWX7LKO mice on ketogenic diet loss weight, but they had the same energy intake, energy expenditure and physical activity (page 16), where does the energy go? For a purpose of transparency, mice body weight (g) as well as food consumption (kcal/day/mouse) should be presented as a timeline. Especially with ketogenic diet, the body weight and food consumption could vary based on the duration of the diet where there is an adaptation period plus ketogenic diet could decrease appetite over time (PMID: 25698989). One data point (Supplementary 6 i) in this case for energy intake is too simplistic to draw any conclusions.

Response: We apologize for the confusion. **Fig. 6a** and **Fig. 6b** are independent studies. For **Fig.6a**, mice were fed a keto diet continuously for 20 days, initiated at 11 weeks of age. To gain more insight into the different phenotypes, we further subjected the 12-week-old mice to the indirect calorimetry study (**Fig. 6b**), wherein mice were fed a normal diet for 2 days in the metabolic cage (day 1 is acclimation, thus data was excluded) followed by acute switching to a keto diet for another three days. We've clarified this in the legend.

Mice body weights were presented as a timeline (Fig. 6a), however the corresponding food consumption (kcal/day/mouse) were not provided due to the butter and soft texture of the keto diet we used (cat. no. S3666; Bio-Serv; Fig for Reviewer 1a below). For the 20-day keto diet study in Fig. 6a, mice were housed three per cage and keto paste was served in a petri dish and replaced every three days. We observed great spill and mixture with bedding of the keto diet during the three-day period, thus making the measurement of calorie intake difficult and inaccurate. For this reason, we did not include the data. For the indirect calorimetry study, keto diet still could not fit in the food hopper of the metabolic cage, thus calorie intake could not be recorded automatically. As shown in the Figure for Reviewer 1b below, food consumption was only recorded during the normal diet feeding, which displayed no significant difference between genotypes. However, during the three-day keto diet indirect calorimetry study, mice were housed individually, and the spill of food was alleviated, making it easier and more accurate to calculate the average food consumption as shown in Supplementary Fig. 6i.

Fig for Reviewer 1a

Fig for Reviewer 1b

Supplementary Fig. 6i

Both body weight control and the effects of keto diet may involve multiple unclear mechanisms. We also could not exclude other effects on body weight control, including nutrient absorption, dehydration, faecal loss, and thermogenesis. We agree with the reviewer that our data is not robust enough to draw a solid conclusion, thus we have softened our language and emphasized that increased weight loss of Fbxw7^{L-/-} mice upon a keto diet was underscored by their abnormal metabolism reflected by significantly blunted ability to activate hepatic FAO as well as peripheral tissue wasting.

Minor point 1. Supplementary 6 k and text on page 16/17. Authors state: “A keto diet triggered liver fat accumulation, especially in Fbxw7LKO, accompanied by.... reduction in hepatic ketogenesis intermediates” but the data that authors present does not show it. AcAc-CoA and HMG-CoA are not reduced in Fbxw7LKO based on the presented data. Authors also have to keep in mind that while both of these metabolites are the substrates for ketones in the mitochondria, they are also contributing to mevalonate pathway and cholesterol production in the cytosol.

Response: We thank the reviewer for pointing this out. After re-normalization to protein content, AcAc-CoA and HMG-CoA showed a nonsignificant trend toward lower expression in Fbxw7-null livers (**Supplementary Fig. 6k**), likely because they are also involved in the mevalonate pathway. We have updated the main text accordingly.

Supplementary Fig. 6k

Minor point 2. Fig 7f and text on page 18. Bdh2 is not robustly involved in mitochondrial ketone body metabolism or ketogenesis. Bdh2 is a cytoplasmic version of Bdh1, with only 20% of sequence identity to BDH1, with a high K_m for ketone bodies (~10 mM), and governs iron homeostasis (PMID: 34633859).

Response: We thank the reviewer for pointing this out. We have modified the main text accordingly.

Reviewer 2:

In this manuscript, Xia and the colleagues investigate the function of liver Fbxw7 in regulating systemic fatty acids homeostasis and contributing to NASH development. Overall, the results are convincing, and the manuscript is well written.

We appreciate that the reviewer recognizes the significance of our study and provided us with very helpful comments. We have carefully addressed all points raised in our revised manuscript. Please see our detailed response below.

1. Please add the line number so it makes the reviewers easy to quote the sentences.

Response: We have added line numbers accordingly.

2. 1c 1d. The author used the overlapped genes for the analysis, but 1) the NASH F4 is very late and severe stage, and the transcriptome at this time point may be composed of compensatory responses and cannot reflect the causal ones, and 2) the mouse and human RNA-seq data are from independent studies and the author also used different criteria to define the DEGs, thus the number of the overlapped genes is very artificial. They should be moved to supplementary, and here in the main figure the mouse liver RNA results should be enough and better.

In addition, the $|FC| > 1.30$ with $p < 0.05$ is too mild to define the DEGs. I suggest at least to use $|FC| > 1.5$ and $FDR < 0.05$, and also GO of up- and down-regulated genes should be separated.

Response: We acknowledge that the human NASH F4 stage is the latest and most severe stage investigated in the study. We agree with the reviewer that this severe stage may be confounded by compensatory responses, however it is always difficult to discern between causal and compensatory mechanisms in any phenotype presented. We have **replaced Fig. 1c** with a **New Supplementary Fig. 1i** to show an increase in overlapping DEGs in livers of Fbxw7 LKO with DEGs from human livers with varying stages of NASH that were compared to either NAFL or NASH F0-1 as a baseline. This new figure, showing increased overlap with late-stage NASH (F4), suggests shared genetic characteristics and connections between decreased Fbxw7 expression/function and NASH progression.

New Supplementary Fig. 1i

Indeed, different criteria were used for identifying DEGs from mouse livers of Fbxw7 LKO mice compared to Flox controls (DEGs; $p < 0.05$ and fold change $> |1.30|$) and NASH-related DEGs in humans (DEGs; $q < 0.05$ and fold change $> |1.50|$). Regarding the above figure, the human NASH-related DEG lists were generated from data on a discovery cohort including $n = 53$ NAFL and $n = 153$ NASH patients. The DEGs identified in Fbxw7 LKO livers stem from $n=3$ Flox controls and

n=3 Fbxw7 LKO mice. Using the more stringent cut-off criteria for DEGs in livers of Fbxw7 LKO as suggested by the reviewer (DEGs; $q < 0.05$ and fold change $> |1.50|$) results in a lower number of overlapping genes with human NASH but consistently shows increased overlap with NASH progression (**Figure for Reviewer 2** below).

Fig for Reviewer 2

DEGs: $q < 0.05$ and fold change $> |1.50|$

Considering the lower number of replicates in the identification of DEGs from Fbxw7 LKO mice and the widely accepted notion that transcriptional changes of individual metabolic genes are small, we opted to use less stringent criteria backed-up by extensive validation by RT-qPCR on a larger sample size.

3. 1f. Fbxw7 should also be included to the WB to show the KO efficiency.

Response: We thank the reviewer for pointing this out. We have now added the detection of Fbxw7 for all the Western blots performed using Fbxw7-null livers, such as **Original Fig. 1f, Now Fig. 1e**.

Fig. 1e

4. 1g 1h. Staining of marks of intrahepatic macrophage, fibroblast and hepatic stellate cells in Fbxw7 Flox and LKO livers should be included, and this would be most direct and solid than the predicted results 1g-1j.

Response: We agree with the reviewer and performed immunohistochemistry (IHC) of macrophage marker Aif1 and hepatic stellate cell/fibroblast marker αSMA, which confirmed elevated immune infiltration of Fbxw7-null livers, see **New Fig. 1j, k**.

5. How the liver weight changes after deletion of Fbxw7?

Response: We have included the liver weight changes (**New Supplementary Fig. 1g**), which demonstrated more than 29% increase upon hepatic deletion of Fbxw7.

New Supplementary Fig. 1g

6. 2g,2h. Whether the expression of SREBP1c (mRNA and protein) changes after deletion of Fbxw7?

Response: We have examined the mRNA and protein level upon the fasting and refeeding transition, when the transcription and proteolytic processing of SREBP-1 are both active and observed negligible differences between genotypes (**New Supplementary Fig. 2i, j**).

New Supplementary Fig. 2i

New Supplementary Fig. 2j

7. 2l. How is the blood insulin level?

Response: We have examined serum insulin levels and observed significantly reduced fed insulin levels of $Fbxw7^{L-/-}$ mice (**New Fig. 2p**), which might contribute to their increased lipolysis and adipose tissue loss.

New Fig. 2p

8. 3a. To better show the quality of the RNAseq from $Fbxw7$ KO and Flox livers, the author should have a PCA plot in supplementary. In addition, it is very misleading if only use $p < 0.05$ to define the significantly changed genes. Please use at least $|FC| > 1.5$ and $FDR < 0.05$ to find DEGs.

Response: We have included a PCA plot for the RNA-seq data (**New Supplementary Fig. 1h**), which showed a substantial difference between genotypes, as well as tight clustering across replicates, indicating low contamination and high reproducibility.

New Supplementary Fig. 1h

As discussed above, all DEGs shown in **Fig. 3a** satisfy the cut-off $p < 0.05$ and fold change $> |1.30$ with detailed changes of all genes presented in the corresponding source data.

9. 3l. Same as 1f, $Fbxw7$ should also be plotted to show the KO efficiency.

Response: We have added the detection of $Fbxw7$ for all the Western blots performed using $Fbxw7$ -null livers as well as livers subjected to the acute loss of $Fbxw7$ such as **Original Fig. 3l**, **Now Fig. 3m**.

Fig. 3m

10. 7a. Since the author has the RNAseq data, why still use RT-PCR results for the heatmap, which is less reliable.

Response: We agree with the reviewer that RNA-seq is regarded as more sensitive and provides “TRUE” overall expression of a gene based on the average expression of all isoforms sequenced. However, RT-qPCR can be advantageous statistically when increasing the number of biological replicates and serves to validate key findings. When performing the RNAseq analysis, we excluded lowly expressed genes with an average read count lower than 10 across all samples.

According to our RNAseq data, the expression of some NASH-related genes is below detection, shown as grey squares in the Z score RNAseq heatmap (**Fig for Reviewer 3**). In general, our qPCR analysis confirmed the RNA-seq. More importantly, we have several conditions analyzed in **Fig. 7a**, while only the fed state has corresponding RNAseq data. In order to prove that metabolic stress exacerbates NASH, we opted to show RT-qPCR data with more replicates.

Fig for Reviewer 3

Fig. 7a

11. 8b. It would be better to add the correlation heatmap with FBXW7 and PPARA besides ESRRA.

Response: Given that Fbxw7 targets multiple factors for degradation, we sought to focus our attention on its downstream effectors for their implication in NASH. Aside from ERR α , we were inclined to add the suggested correlation heatmap for PPAR α . However, to our surprise we did not observe an anticipated negative correlation between NASH-related genes and PPAR α transcript levels. Upon closer examination of the 40 genes related to NASH, inflammation, fibrogenesis or ER stress & apoptosis (Fig 8a, b), only 6 were identified as direct targets of PPAR α by ChIP-seq in mouse liver whereas 33 were identified as direct targets of ERR α by ChIP-seq in Fbxw7-null livers (binding sites +/- 20kb TSS), in large support of why we were able to achieve a significant positive correlation with ERR α expression. Given these observations, we did not show the correlation heatmap with PPAR α . The contribution of PPAR α loss-of-function to NASH in Fbxw7-null liver is most likely tied to its regulation of other target genes, most notably FAO genes, which are targeted by ERR α for competition with PPAR α for binding to composite elements (New Fig. 8j and New Supplementary Fig. 1m).

12. 8k. Remove the motifs figure as well as the related sentences in the main text since they are wrong. These two ERRE are actually both half NR binding site, while the top PPRE seems not any NR motif and the bottom PPRE is actually a full NR (DR1) binding (which can be the motif of any DR1 NR).

Response: We thank the reviewer for noting this grave mistake. Indeed, the two motifs shown (ERRE and PPRE) were not correct, a momentous error that we realized ourselves almost as soon as the initial submission was completed but already sent for reviews. We were not happy. Based on an entirely new and more complete analysis of the ChIP-seq data, we have now updated our findings presented in New Fig. 8j. First, following intersection of PPAR α ChIP-seq targets with ERR α ChIP-seq targets in either Flox or Fbxw7-null livers (binding sites +/- 20kb TSS), we now indicate whether PPAR α and ERR α display co-occupancy or distinct genomic binding at the shared target genes. Second, motif instances for ERREs and PPREs based on the motifs defined in HOMER for all ERR and PPAR isoforms were determined on the co-occupied genomic regions between PPAR α and ERR α in Fbxw7-null livers. This analysis identified the presence of both an ERRE and PPRE in 42% of overlapping peaks of which 73% were defined as a hybrid ERRE/PPRE motif (overlapping elements) predominantly manifesting as the defined ERR α motif CAAAGGTCA embedded within the PPAR α DR1 motif AGGTCAAAGGTCA. The new data presented in New Fig. 8j increases the impact of our findings and largely supports competition of both factors at a subset of shared target genes bearing composite elements.

New Fig. 8j

Minor point 1. 1 m&n, Usually the over-expressed protein level is much higher than the physiological level, so this result makes no big sense for me. Although it fits your hypothesis, I suggest to move it to supplementary figs.

Response: As suggested, we have moved the **original Fig. 1m, n** to **Supplementary Fig. 1o, p**.

Minor point 2. 3a. Please include the number of mice/livers used for RNAseq.

Response: We have stated the number of mice (n = 3) used for RNAseq in the corresponding legend.

Minor point 3. 4a. Please also indicate the number of genes identified in each GO CC.

Response: The number of genes is now shown for each GO analysis.

Reviewer 3:

This is a very elegant and highly comprehensive work on the role of FBXW7 in the development of pathogenesis of NAFLD and NASH using a combination of diverse approaches. They found that reduced levels of FBXW7 and PPARA as a key feature of advanced NASH. Interestingly, they demonstrated that loss of Fbxw7 in hepatocyte downregulates the almost entire fat degradation network without affecting the lipogenic program, which is critical for NAFLD initiation but is unlikely to be the main contributor to NASH progression. Their results convincingly show that the accumulated TG in Fbxw7-null livers is predominantly caused by their massive defects in fat clearance associated with attenuated PPARA expression. NASH. They further show that inhibition of FBXW7 substrate ERRA alleviates NASH via competition for the same genomic binding sites that recruits PPARA. Overall, this study suggest that simultaneous PPARA activation and ERRA inhibition as a potential therapeutic NASH treatment.

While this study is comprehensive, it is overall too descriptive and lacks mechanistic insights. Most importantly, while PPARA is a key player in the action of FBXW7, the molecular details on how FBXW7 controls the activity of PPARA either directly or indirectly are completely missing. Additional experiments showing this missing link will make this work stronger and more insightful and complete.

With adding/addressing the major mechanistic studies as suggested above, this reviewer recommends publication of this beautiful work in Nature Communications.

We thank the reviewer for recognizing the biological significance of our study and constructive comments. We have carefully addressed the points raised in our revised manuscript notably by demonstrating that Fbxw7 control of PPAR α occurs in an independent manner via ERRA transcriptional regulation. Please see our detailed response below.

Response: We first proved that impaired hepatic fat catabolism in Fbxw7^{L-/-} mice was directly caused by loss of Fbxw7 rather than long-term compensation, as acute Fbxw7 depletion via injection of Flox mice with adenovirus expressing control or CRE recombinase led to significant reductions in mRNA and protein levels of hepatic PPAR α and FAO genes (**Fig. 3l, m** and **Supplementary Fig. 3j**). Furthermore, both transient and stable deletion of FBXW7 in HepG2 cells resulted in remarkable attenuation of PPAR α mRNA and protein levels (**New Supplementary Fig. 3k-n**). Consistently, both transient and stable overexpression of FBXW7 in HepG2 cells upregulated PPAR α transcripts and protein abundance accompanied by augmented FAO gene expression, with the induction being more evident upon stable FBXW7 overexpression (**New Supplementary Fig. 3o-s**). Interestingly, although FBXW7 stimulates PPAR α in a cell-autonomous manner, immunoprecipitation (IP) experiments demonstrated no physiological interaction between endogenous Fbxw7 and PPAR α in mouse liver (**New Supplementary Fig. 3t**). This observation was further supported by IP studies performed in HepG2 cells treated with proteasome inhibitor MG132 as well as in 293T cells overexpressing FBXW7 and PPAR α (**New Supplementary Fig. 3u, v**), implying indirect regulation of PPAR α by FBXW7.

New Figs in Supplementary Fig. 3

The downregulation of PPAR α transcription in hepatocytes lacking Fbxw7 (Fig. 3f, k-m and Supplementary Fig. 3k-n) together with increased ERR α recruitment to the PPAR α locus in Fbxw7^{L-/-} mice (New Supplementary Fig. 8a), suggests that Fbxw7-mediated regulation of PPAR α is ERR α -dependent. In line with this notion, both genetic and pharmacological inhibition of ERR α in HepG2 cells abrogated the observed reduction in PPAR α mRNA and protein expression upon FBXW7 deletion (New Supplementary Fig. 8b-f). Consequently, suppressed expression of mRNA and proteins of essential genes involved in hepatic FAO were partially restored *in vivo* by C29-mediated ERR α inhibition in Fbxw7^{L-/-} mice, in line with augmented hepatic FAO activity (Fig. 8k and New Supplementary Fig. 8g, h).

Together, our new findings suggest that Fbxw7 indirectly promotes *PPARA*, at least partly through ERR α inhibition.

New Supplementary Fig. 8

REVIEWER COMMENTS

Reviewer #2 (Remarks to the Author):

While the revised manuscript has shown vast improvement with the newly obtained results, and many of my previous concerns have been addressed, I am still somewhat skeptical about the proposed mechanism based on the limited molecular experiments presented in the study. Please consider the following remaining comments.

Major:

Line 443-447 Fig 8g:

Could you please explain how you define the fold change of ERRa binding since there is only 1 ChIP-seq was performed in each genotype?

Line 458-459 Fig 8 j:

To obtain a clearer understanding of dynamic ERRa and PPARa binding in the presence and absence of Fbxw7, it would be ideal to perform ERRa and PPARa ChIP-seq on the same liver samples with a minimum of three replicates. This will ensure more robust and reliable results, and allow for a more confident interpretation of the mechanism at molecular level.

If it is not feasible, at least a higher-quality PPARa ChIP-seq dataset with a comparable number of peaks to ERRa should be utilized. The selected PPARa ChIP-seq dataset had suboptimal quality, and the difference of peak number will lead to the totally different conclusion.

It is important to note that without PPARa binding information in Fbxw7L^{-/-} livers, the identification of functional PPARa bindings (those that change in Fbxw7L^{-/-} compared to control) is impossible. Consequently, the dissection of ERRa and PPARa bindings in these co-occupied loci would be meaningless.

Fig 9, Supp 7i, line 1411:

I recommend including additional snapshots of ERRa bindings on PPARa target genes to provide a more comprehensive understanding of the regulatory interactions between these two transcription factors.

Minor:

Line 458-459, Fig 8 j:

I would recommend to remap the raw reads from fastq files to the mm10 reference genome instead of relying on lift-over of peaks from mm9. This ensures more accurate alignment and improve the reliability of downstream analyses.

Supp 7i, Supp 8a,

When showing the ChIPseq tracks, it is important to include negative regions where no significant changes in ERRs bindings are observed in adjacent genes, in addition to positive regions.

Line 440-443, Supp 7g:

I am slightly concerned about such a big difference in the number of ERRa peaks between control and Fbxw7L^{-/-} livers. To address this issue, I suggest using a mixture of the two inputs for the peak calling, as well as exploring other peak calling methods such as Homer Findpeaks. This will enable us to determine whether the observed changes in ERRa binding are consistent across different peak calling methods, especially when you have only one ChIPseq.

Reviewer #3 (Remarks to the Author):

The authors nicely clarified the major issues raised by this reviewer, demonstrating the impact of FBXW7 in NASH resulting from its ability to control ERRA levels.

REVIEWER COMMENTS

Reviewer #2 (Remarks to the Author):

While the revised manuscript has shown vast improvement with the newly obtained results, and many of my previous concerns have been addressed, I am still somewhat skeptical about the proposed mechanism based on the limited molecular experiments presented in the study. Please consider the following remaining comments.

Major:

1) Line 443-447 Fig 8g: Could you please explain how you define the fold change of ERR α binding since there is only 1 ChIP-seq was performed in each genotype?

Response: Actually, Fig. 8g is showing the log₁₀ p-value and log₂ fold-changes of genes from RNA-seq data. Among the significant DEGs in Fbxw7-null livers ($p < 0.05$, $|FC| > 1.30$), genes in blue exhibited no differential binding by ERR α from ChIPseq data whereas genes in purple exhibited differential ERR α ChIPseq binding between Flox and Fbxw7-null livers. All other genes in black are genes from the RNA-seq that did not meet the criteria for differential expression ($p < 0.05$, $|FC| > 1.30$). The differential ERR α binding observed by ChIP-seq was indeed based on an n=1 experiment from individually pooled ChIPs to minimize technical error. Differential binding here was defined as a differential peak intensity difference ($|FC| \geq 1.5$) in ChIP-seq signal intensity (FPKM+1) between Fbxw7-null and Flox liver restricted to peaks found within ± 20 kb of gene TSSs (this was clarified in the manuscript). Although we cannot perform statistical analyses on the differential binding observed for ERR α as the ChIP-seq experiments were performed in single replicate, the increased ERR α protein levels in Fbxw7-null liver support the observed general increase in ChIP-seq binding intensity and underly the larger number of identified binding events.

2) Line 458-459 Fig 8 j:

To obtain a clearer understanding of dynamic ERR α and PPAR α binding in the presence and absence of Fbxw7, it would be ideal to perform ERR α and PPAR α ChIP-seq on the same liver samples with a minimum of three replicates. This will ensure more robust and reliable results, and allow for a more confident interpretation of the mechanism at molecular level.

If it is not feasible, at least a higher-quality PPAR α ChIP-seq dataset with a comparable number of peaks to ERR α should be utilized. The selected PPAR α ChIP-seq dataset had suboptimal quality, and the difference of peak number will lead to the totally different conclusion.

It is important to note that without PPAR α binding information in Fbxw7L $^{-/-}$ livers, the identification of functional PPAR α bindings (those that change in Fbxw7L $^{-/-}$ compared to control) is impossible. Consequently, the dissection of ERR α and PPAR α bindings in these co-occupied loci would be meaningless.

Response: As we no longer house the Fbxw7-null mouse model, performing de novo triplicate ERR α and PPAR α ChIP-seq experiments on mouse liver is not possible. Alternatively, as the reviewer suggested, a cross-comparison of our ERR α ChIP-seq profiles with a higher-quality PPAR α ChIP-seq dataset would be needed to ascertain the findings presented. To select the best PPAR α ChIP-seq dataset, we screened the ChIP-seq repository ChIP-Atlas which processes public ChIP-seq datasets through the same pipeline facilitating comparison of available ChIP-seq profiles. Based on our screening for PPAR α ChIP-seq datasets performed on liver of WT or control mice, 3 PPAR α datasets were selected for further analysis based on

the total number of called peaks by the CHIP-Atlas pipeline as an indicator of the strength of the ChIPs. The PPAR α dataset that was initially used in the manuscript was maintained as well as two other PPAR α datasets:

1. PPAR α ChIPseq: GSM1514930 (Liver of WT male C57BL/6J treated with PPAR α agonist GW7647; anti-PPAR α sc-9000x, Santa-Cruz); PMID: 25383539.
2. PPAR α ChIPseq: GSM3517890-92 triplicate experiment (Gps2 WT Liver of male C57BL/6; anti-PPAR α MAB3890, Millipore); PMID: 30975991.
3. PPAR α ChIPseq: GSM864671 (Liver of WT female C57BL/6; anti-PPAR α sc-9000, Santa-Cruz); PMID: 22158963. This dataset was initially used in our manuscript.

Next, as per minor point 1, these 3 selected PPAR α ChIPseq datasets were re-analyzed from scratch using raw fastq files for alignment to mm10 and peaks were called using the same ChIP-seq pipeline used for our ERR α ChIP-seq datasets (MACS2 FDR 0.05 with subsequent filtering to FDR 0.001). For full assurance, the ERR α ChIPseq datasets were re-analyzed from scratch in parallel yielding negligible discrepancies as expected stemming from random assignment of reads mapped to multiple loci and removal of peaks from blacklisted and uninformative regions. As per minor point 3, peaks were called using two methods: MACS2 and HOMER findPeaks for both ERR α and PPAR α ChIPseq datasets to ensure that the increased occupancy of ERR α at PPAR α sites observed in Fbxw7-null vs control liver is not bias towards a particular peak calling method. The results from this comprehensive analysis are summarized in Figure 1 – for reviewer below.

Figure 1 - for reviewer

Peak calling: HOMER FDR 0.05

Peak calling: MACS2 FDR 0.001

As the reviewer can fully appreciate, we observed a similar increase in the overlap of PPAR α and ERR α targets (peaks +/- 20kb of gene TSSs) when ERR α was ChIP'd from Fbxw7-null liver, importantly regardless of the PPAR α ChIP-seq selected and independently of the peak calling method employed. Among notable observations, re-analysis of PPAR α dataset 3 (GSM864671) from scratch and direct alignment to mm10 greatly increased the number of PPAR α targets identified using the same pipeline that was reported in the manuscript, thus being more comparable to the ERR α datasets, but without altering the trend in ERR α binding (Figure 1c above vs Figure 8j in manuscript). The number of PPAR α targets identified in dataset 3 (GSM864671) were comparable to those found for dataset 1 (GSM1514930). For PPAR α dataset 2 (GSM3517890-92), the number of targets found were much lower than the other two PPAR α datasets given that triplicate experiments were available, and we opted for a high-confidence list of PPAR α peaks by restricting the peaks to those found common to all three replicates using the HOMER merge peaks tool (-d300). Nevertheless, this robust list of PPAR α peaks resulted in the same trend for ERR α binding. When peaks were called using another peak calling program HOMER findPeaks with default conditions (FDR 0.05), there was a notable reduction in ERR α peaks called, unlike the PPAR α datasets, likely due to weaker binding intensities, but nevertheless we observed the same trend for ERR α binding, exhibiting increased recruitment to PPAR α sites when ERR α was ChIP'd in Fbxw7-null liver vs Flox control (Figure 1d-f above). In fact, this trend in ERR α binding was more evident with HOMER findPeaks. Considering the large difference in peaks called for ERR α between MACS2 (Figure 1a-c above) and HOMER findPeaks (Figure 1d-f above), we performed another analysis with MACS2 with increased stringency by calling peaks with FDR 0.001 (Figure 1g-h above), which generated comparable results to that obtained with HOMER findPeaks (FDR 0.05). We were surprised by the large difference in MACS2 peak calling between 1) setting an FDR alpha to 0.05 with subsequent filtering of peaks to FDR 0.001 (Figure 1a-c above) vs setting the FDR alpha to 0.001 (Figure 1g-i above). The initial FDR alpha setting changes how the independent filtering is performed and thus affects the adjusted p-values attributed. Focusing on the data presented in Figure 1g-i above with the more stringent MACS2 FDR alpha, ERR α and PPAR α shared target genes increased by 6-fold when ERR α was ChIP'd in Fbxw7-null vs Flox liver, but strikingly the number of co-targeted genes with overlapping ERR α and PPAR α peaks (shared binding sites) increased by 30-fold, evidently not a random effect. Importantly, even with a more relaxed MACS2 peak calling method employed in Figure 1a-c above which largely increased the number of peaks detected, particularly for ERR α in Flox liver, we observed the same 5-fold increase in ERR α occupancy at PPAR α sites within co-targeted genes in Fbxw7-null liver which cannot be explained solely by the enhanced repertoire of identified ERR α targets, thus emphasizing the specific context-dependent recruitment of ERR α to PPAR α sites.

Overall, this exhaustive bioinformatics exercise implicating 3 different PPAR α ChIPseq datasets from independent studies involving the use of two different PPAR α antibodies clearly rules out the possibility that the conclusions drawn in the manuscript underly the use of a sub-optimal PPAR α ChIP-seq dataset. The increased occupancy of ERR α at PPAR α -targeted loci in Fbxw7-null liver was reproducible in all analyses, independent of peak calling method, FDR cut-off, and PPAR α ChIP-seq dataset selected, demonstrating the validity of our findings which cannot have arisen by chance alone. We are confident with our study revealing that in Fbxw7-null liver which features a NASH phenotype with increased ERR α protein levels in combination with decreased PPAR α protein levels, that ERR α binding to DNA is increased as well as its recruitment to known PPAR α -bound regions, the latter favored by direct ERR α -mediated attenuation of PPAR α transcription.

After careful review of our findings presented in Figure 1 – for reviewer above, we made the decision to switch from the use of PPAR α ChIPseq dataset 3 to dataset 1 with maintenance of our peak calling settings of MACS2 FDR alpha 0.05 filtered to FDR 0.001 used originally in the manuscript (Figure 1a above). This decision was based on the following:

a) PPAR α ChIP'd signal intensities from dataset 1 were generally stronger than the other two datasets explored. Indeed, dataset 1 made use of a PPAR α agonist. Also, our ERR α ChIP-seqs were performed using male livers and PPAR α dataset 1 involved livers from male mice as opposed to female livers used in dataset 3.

b) While results with the stringent MACS2 FDR threshold of FDR 0.001 was more consistent with the HOMER findPeaks default method (FDR 0.05), we maintained our original MACS2 settings of FDR 0.05 (default) with additional filtering to FDR 0.001 – this was also made clearer in the methods section. Our cut-off is already more stringent than the default MACS2 FDR of 0.05 setting typically used and increasing the stringency further ultimately results in a significant loss in peaks called in the Flox control liver at known established ERR α targets.

The manuscript was updated accordingly and a new updated version of Figure 8j is shown below as well as an updated version of associated Fig. S7m along with new Fig. S7l showing a heatmap representation of ERR α and PPAR α ChIP-seq binding intensities for the 4,056 ERR α /PPAR α overlapping peaks targeting 3,140 shared genes identified through cross-examination of PPAR α liver ChIP-seq (GSM1514930) and our ERR α ChIP-seq in Fbxw7-null liver.

As the reviewer can see in the new updated versions of Figure 8j and Figure S7m above, using a completely new PPAR α ChIP-seq dataset analyzed from raw fastq files using the same pipeline as for ERR α with peaks called using MACS2 FDR 0.05 with filtering to FDR 0.001, the conclusions remain.

We strongly believe that the dissection of shared ERR α /PPAR α binding sites in our study is both warranted and intriguing and this framework sets the stage for more in-depth future investigation. Through ChIP-qPCR assays using the same leftover chromatin pools used for ERR α ChIP-seq study, we now provide evidence for increased ERR α occupancy with a simultaneous decline in PPAR α binding at several co-targeted loci in Fbxw7-null vs Flox liver (Supplementary Fig. 7n).

Of 6 loci examined, 5 showed a differential binding of $|FC| \geq 1.5$ for both ERR α and PPAR α . While these results are limited to one experiment but importantly use the same batch of chromatin, they provide direct evidence for the context-dependent inverse shift in ERR α and PPAR α binding at co-targeted sites dictated at least in part by their altered abundance in Fbxw7-null liver.

Taking into consideration the changes mentioned above, we have updated the text as follows:

“Massive impairment of FAO in Fbxw7-null livers might be driven by competition for DNA binding between redundant ERR α and drastically reduced PPAR α protein. To investigate this possibility, we compared our ERR α ChIP-seq data with an available liver PPAR α ChIP-seq dataset (GSM1514930) re-analyzed from raw fastq sequencing files using the same pipeline for ERR α . Noteworthy, the selected PPAR α ChIP-seq dataset was performed on livers of mice treated with PPAR α agonist GW7647, thus ensuring a strong catalogue of PPAR α -targeted genes. Examination of ERR α occupancy of PPAR α target genes revealed a 64% overlap in Flox livers, with this percentage rising to 90% in Fbxw7-null livers (Fig. 8j). The augmentation in PPAR α /ERR α co-targeted genes owing to Fbxw7 loss was largely ascribed to ERR α enrichment at established PPAR α binding sites (Fig. 8j), signifying a context-dependent recruitment of ERR α to PPAR α sites. Consensus ERR (ERRE) and PPAR (PPRE) response elements were identified in 38% of overlapping peaks (1,552 of 4,056), the majority (74%) constituting a hybrid ERRE/PPRE element predominantly manifesting as the ERR α motif CAAAGGTCA embedded within the PPAR α DR1 motif AGGTCAAAGGTCA, reinforcing the potential competition between ERR α and PPAR α (Fig. 8j and Supplementary Fig. 7l). A

subset of *Fbxw7*-null liver DEGs marked by co-targeted loci harboring consensus ERRE and PPRE elements were found highly enriched in FAO (Supplementary Fig. 7m). ChIP-qPCR analyses performed on the same chromatin pools used for *ERRα* ChIP-seq study confirmed increased *ERRα* occupancy at several examined genes in *Fbxw7*-null versus *Flox* liver with a simultaneous decline in *PPARα* binding (Supplementary Fig. 7n). Together, these findings support that competition for occupancy between *ERRα* and *PPARα* indeed exists whereby attenuated *PPARα* activity in NASH would favor binding of *ERRα* to a specific set of genes.”

3) Fig 9, Supp 7i, line 1411:I recommend including additional snapshots of *ERRα* bindings on *PPARα* target genes to provide a more comprehensive understanding of the regulatory interactions between these two transcription factors.

Response:

We have indirectly addressed this point in response to major point 2. In response to this comment, we have removed the original Supplementary Fig. 7i as the binding profiles shown were for genes that were targeted by *ERRα* but not necessarily by *PPARα*. We now show binding profiles for 6 strong *ERRα*/*PPARα* co-targeted loci at genes found deregulated in *Fbxw7*-null liver and mostly associated with FAO (Supplementary Fig. 7n). We now also provide associated ChIP-qPCR assays performed on the same chromatin pools showing differential binding of *ERRα* and *PPARα* in *Fbxw7*-null vs *Flox* liver consistent with the upregulation of *ERRα* and downregulation of *PPARα* levels observed in *Fbxw7*-null liver (Supplementary Fig. 7n). These results support the model presented in Figure 9.

Minor:

1) Line 458-459, Fig 8 j: I would recommend to remap the raw reads from fastq files to the mm10 reference genome instead of relying on lift-over of peaks from mm9. This ensures more accurate alignment and improve the reliability of downstream analyses.

Response: This was done. Please refer to our response to major point 2. Again, we thank the reviewer for this comment as re-analysis of *PPARα* ChIP-seq data from raw fastq files with alignment to mm10 and use of the same peak calling method employed for that of *ERRα* greatly improved the quality and reliability of the downstream analyses.

2) Supp 7i, Supp 8a, when showing the ChIPseq tracks, it is important to include negative regions where no significant changes in ERRs bindings are observed in adjacent genes, in addition to positive regions.

Response: While keeping in mind that ERR α may in fact bind to genes adjacent to the gene of interest, we have updated Supplementary Fig. 8a to show additional genes in the vicinity of the *PPARA* locus. Also, as mentioned in response to major point 2, we have removed Supplementary Fig. 7i and focused our attention on genome browser views of ERR α and PPAR α binding at co-targeted loci in Supplementary Fig. 7n. For the 6 co-targeted genes shown, we have made sure to show broader genome views (10 kb regions) with a clear visual overlap in ERR α and PPAR α binding.

3) Line 440-443, Supp 7g: I am slightly concerned about such a big difference in the number of ERR α peaks between control and Fbxw7^{L-/-} livers. To address this issue, I suggest using a mixture of the two inputs for the peak calling, as well as exploring other peak calling methods such as Homer Findpeaks. This will enable us to determine whether the observed changes in ERR α binding are consistent across different peak calling methods, especially when you have only one ChIPseq.

Response: As requested, the proposed exploration of changes in ERR α binding using two peak calling methods has been formally addressed in major point 2. Both MACS2 and HOMER findPeaks resulted in a large increase in number of peaks called in Fbxw7-null vs control liver, supported by the augmented levels of ERR α protein in the absence of Fbxw7. This difference in ERR α peaks became more apparent with increasing stringency of the MACS2 peak calling threshold employed.

As per the demand of the reviewer, we have also addressed whether the increased number of peaks called for ERR α in Fbxw7-null vs control liver was influenced by the inputs used as control. To this end, ERR α peaks called using respective input controls were compared to ERR α peaks called after correction to both input controls. For the latter, peaks were called using each input separately and only peaks found common to each correction were retained using the Homer merge peaks tool (-d300). As shown below in Figure 2 – for reviewer, correction to both inputs had no bearing on the significant difference in ERR α peaks identified between Fbxw7-null and control liver using either MACS2 or HOMER FindPeaks. In fact, correcting to both inputs, being more stringent than correction to one input, reduced the number of peaks

called in Flox liver to a greater extent than that in Fbxw7-null liver, thus amplifying the difference in peaks between the genotypes. Given the nature of NASH exhibited in Fbxw7-null liver, we do not feel it is biologically relevant to correct the peaks called using both inputs and have retained our analysis to ERR α peaks called in Fbxw7-null or Flox liver with correction using respective genotype input controls.

Figure 2 - for reviewer